# Engineered *Lactococcus lactis* secreting Flt3L and OX40 ligand for in situ vaccination-based cancer immunotherapy

Junmeng Zhu[1,3], Yaohua Ke[1,3], Qin Liu[1,3], Ju Yang[1], Fangcen Liu[2], Ruihan Xu[1], Hang Zhou[1], Aoxing Chen[1], Jie Xiao[1], Fanyan Meng[1], Lixia Yu[1], Rutian Li[1], Jia Wei[1] & Baorui Liu ®[1] ✉

In situ vaccination is a promising strategy to convert the immunosuppressive tumor microenvironment into an immunostimulatory one with limited systemic exposure and side effect. However, sustained clinical benefits require long-term and multidimensional immune activation including innate and adaptive immunity. Here, we develop a probiotic food-grade *Lactococcus lactis*-based in situ vaccination (FOLactis) expressing a fusion protein of Fms-like tyrosine kinase 3 ligand and co-stimulator OX40 ligand. Intratumoural delivery of FOLactis contributes to local retention and sustained release of therapeutics to thoroughly modulate key components of the antitumour immune response, such as activation of natural killer cells, cytotoxic T lymphocytes, and conventional-type-1-dendritic cells in the tumors and tumor-draining lymph nodes. In addition, intratumoural administration of FOLactis induces a more robust tumor antigen-specific immune response and superior systemic antitumour efficacy in multiple poorly immune cell-infiltrated and anti-PD1-resistant tumors. Specific depletion of different immune cells reveals that CD8+ T and natural killer cells are crucial to the in situ vaccine-elicited tumor regression. Our results confirm that FOLactis displays an enhanced antitumour immunity and successfully converts the 'cold' tumors to 'hot' tumors.

Intratumoural (i.t.) administration of cytokines and antibodies is an essential strategy of in situ vaccination (ISV), aiming to generate 'in situ antigen factories' with higher local bioavailability and lower systemic toxicity[1]. However, it is difficult to ensure the persistence of the locally delivered small-molecule drugs because of their fast metabolism via the tumor vasculature and/or lymphatics[2]. For example, intratumoural administration of Fms-like tyrosine kinase 3 ligand (Flt3L) or OX40 agonists has been applied in clinical trials for hematologic malignancies and some solid tumors but obtained few benefits[3,4]. The rapid degration limited the efficacy of free Flt3L or OX40 agonists and it was

critical to develop an efficient carrier that acted as immune adjuvants to get the most out of the drugs. Compared to other biomaterials such as polymeric microspheres[5], microneedle patches[6] and injectable hydrogels[7], probiotics can be natural vectors and at the same time activate innate and adaptive immunity.

In recent years, the rapid developments of synthetic biology have enabled the design of intelligent microbial delivery systems for therapeutic applications[8]. To be more specific, bacteria can be engineered into safe, targeted and immune-boosting drug carriers[9]. Among them, probiotics such as lactic acid bacteria (LAB) have been found in the

---

[1]The Comprehensive Cancer Centre of Nanjing Drum Tower Hospital, The Affiliated Hospital of Nanjing University Medical School, 321 Zhongshan Road, Nanjing 210008, China. [2]Department of Pathology, The Affiliated Hospital of Nanjing University Medical School, 321 Zhongshan Road, Nanjing 210008, China. [3]These authors contributed equally: Junmeng Zhu, Yaohua Ke, Qin Liu. ✉e-mail: baoruiliu@nju.edu.cn

human gut and positively affect immunotherapy[10–12]. The unique feature makes these strains attractive for clinical use without further attenuation.

Probiotics are linked to an enhanced immune response to cancer immunotherapy. Firstly, immune surveillance and a hypoxic environment in the necrotic core of tumors provide a critical niche for some anaerobic or facultative anaerobic bacteria to deeply colonize[9,13]. Meanwhile, several probiotics have been reported to effectively mobilize the immune system due to their abundance in pathogen-associated molecular patterns (PAMPs) such as lipopolysaccharide (LPS) and peptidoglycan. Peptidoglycan, widely existing in gram-positive bacteria, can be a nonspecific agonist for the immune response. Different peptidoglycan fragments sensed by various pattern recognition receptors (PRRs), including C-type lectin receptors (CLRs), toll-like receptors (TLRs), NOD-like receptors (NLRs), AIM2-like receptors (ALRs), RIG-I-like receptors (RLRs), peptidoglycan binding proteins (PGBPs), etc., will further promote innate and adaptive immune response through activation of transcription factors such as nuclear factor κB (NF-κB) and mitogen-activated protein kinase (MAPK) phosphorylation cascades, etc[14]. Given safety and practicality, food-grade *Lactococcus lactis* has unique advantages among probiotics in delivering drugs[15] and has been tested in patients as a single agent or in combination with other therapeutic modalities (NCT04048174, NCT04760353, NCT03751007, NCT04997057 and NCT03893162) over the past decades. As a facultative anaerobe, *Lactococcus lactis* can also target and localize within the hypoxic tumor microenvironment (TME)[16]. Besides, the immunomodulatory effects of *Lactococcus lactis* on dendritic cells (DCs), natural killer (NK) cells and macrophages have been demonstrated via MyD88-dependent manner or NF-κB signaling pathway[17,18].

In this work, we design a bifunctional engineered *Lactococcus lactis* (FOLactis) delivering an encoded fusion protein of Flt3L and co-stimulator OX40 ligand (OX40L). Injection(i.t.) of FOLactis can kill cancer cells by direct cell lysis and induce strong anti-bacteria immune responses, releasing large amounts of tumor-specific antigens[19–22]. These antigens consequently couple with the bacteria-associated danger signaling and promote antigen cross-presentation by DCs in TME and tumor-draining lymphnodes (TDLNs). Flt3L-expressing FOLactis can directly expand the local proliferation and differentiation of a specialized cross-presenting DC subset, conventional-type-1-dendritic cells (cDC1) especially CD103+ CD11c+ and CD8α+ CD11c+ dendritic cells, which are professional at taking up dead tumor cells, transporting tumor-specific antigens, promoting endogenous anti-tumour T cell epitope spreading[23] and re-stimulating tumor-specific CD8+ T cells[24]. Moreover, TLRs agonists have been proved to synergize with OX40 signaling[25–28]. We hypothesize that *Lactococcus lactis* can increase the expression of OX40 on CD4+ T cells in TME by PAMPs[26] and further maximize the effect of OX40L, promoting tumor-infiltrating effector T cells ($T_{eff}$) activation while inhibiting regulatory T cells ($T_{reg}$) function[29] (Fig. 1a). Using multiple subcutaneous and orthotopic mouse models with poor immune cell infiltration and anti-PD1-resistance, our study indicates that the bifunctional FOLactis leads to a significant tumor regression mainly by increasing the number of cDC1 and restoring the cytotoxic T lymphocytes (CTLs) response in TME. On top of it, this ISV can exert an abscopal effect and synergize with an anti-PD1 antibody, converting so-called immune-desert or "cold" phenotype into "hot" inflamed tumors. Altogether, we find that *Lactococcus lactis* has the potential to be a platform for delivering immunologic adjuvants and providing long-term protection against tumor rechallenges, with little harm to other organs.

## Results
### Construction and characterization of Flt3L-OX40L fusion protein (FO) expressed by *Escherichia coli* or *Lactococcus lactis*
For purified FO production, the extracellular domain of the Flt3L at the N-terminus was fused to the extracellular domain of the OX40L at the C-terminus by a matrix metalloproteinase (MMP)-sensitive peptide Pro-Val-Gly-Leu-Iso-Gly (PVGLIG)[30]. And MMP is most abundant in the tumor extracellular matrix. Then, FO gene was successfully inserted into the pET28a vector (Supplementary Fig. 2a). After the induction with isopropyl β-D-1-thiogalactopyranoside (IPTG), FO was expressed in *Escherichia coli* BL21 (DE3) and refolded from inclusion bodies, migrating at approximately 37 kDa in SDS-PAGE (Supplementary Fig. 2b). According to the manufacturer's instructions, the refolded FO was further purified using metal affinity chromatography under native conditions using an AKTA system HisTrap HP column (GE healthcare, CT, USA) (Supplementary Fig. 2d). Purified fractions were collected and verified by western blotting (WB) analysis, confirming the expected molecular size (Supplementary Fig. 2c).

To generate FOLactis, we used a Lactis-E. coli shuttle vector pNZ8148 (Supplementary Fig. 1a). FOLactis was induced by nisin and then subjected to WB analysis, further confirming the molecular size. We found enhanced protein bands in lysate samples of FOLactis with molecular weights of approximately 37 kDa (Fig. 1b, lane 1) compared to the sample before induction (Fig. 1b, lane 2) and lysate samples of wild-type (WT) Lactis (Lactis) cloned into pNZ8148 vector after induction (Fig. 1b, lane 0). Murine Flt3L and murine OX40L fusion proteins in bacterial cell lysates were detected by ELISA against murine Flt3L and murine OX40L, respectively (Fig. 1c). The 37 kDa protein was analyzed by mass spectrometry (MS) and the results showed the successful production of FO (Fig. 1d, Supplementary Fig. 1b, c, Supplementary Data 1, and Supplementary Data 2). To further confirm whether PVGLIG could be cleaved by matrix metalloproteinases (MMPs), we extracted and purified FO from the lysates of FOLactis. After treatment with collagenase for five hours, most FO can be cleaved, as shown in the SDS-PAGE images (Supplementary Fig. 3). Therefore, we believed that FO could be cleaved by MMPs, which was highly expressed in tumors, then Flt3L and OX40L would play their roles in TME, respectively.

### FO and FOLactis efficiently induced the activation of DCs and T cells in vitro
Antigen recognition and internalization by DCs is the critical step to process and present tumor-specific antigens[31]. Therefore, we examined the internalization of FOLactis by bone marrow-derived DCs (BMDCs). We successfully designed an engineered Lactis expressing a fusion protein composed of Flt3L, OX40L and sfGFP (FOLactis-sfGFP), indicating that the expressed heterologous protein retained its biological structure and activity (Supplementary Fig. 4a). FOLactis dyed with DiO or FOLactis-sfGFP were incubated with BMDCs at a ratio of 10:1 for two hours. Most BMDCs internalized these engineered Lactis and trapped them within lysosomes (Fig. 2a and Supplementary Fig. 4b, d). When bacteria invade the body, they are attacked by immune system via three main ways: (i) via complement-mediated lysis[32,33]; (ii) via phagocytosis[34–36]; (iii) via cell-mediated immunity[37]. According to these studies, we assumed that the fusion protein would be released as FOLactis died. Notably, FOLactis-sfGFP released the fusion protein in PBS slowly and continuously, while it took lesser time to release the fusion protein in PBS mixed with BMDCs or splenocytes (Supplementary Fig. 4c, e, f). We further comfirmed the activity of the fusion protein (Flt3L-OX40L-sfGFP) by immunofluorescence (Fig. 2h–i).

To evaluate the DCs activation, different concentrations of Lactis or FOLactis were added to BMDCs in vitro for 24 h. Normal saline (NS) and LPS were used as negative and positive controls, respectively. We found that the activation effects of FOLactis on BMDCs were dose-dependent. There was a significant increase in the percentage of mature DCs (CD80+ CD86+) in the FOLactis group within the concentration range of $10^7$–$10^8$ CFU ml$^{-1}$, even much better than the LPS group (24.5%). When the concentration of Lactis reached $10^7$ CFU ml$^{-1}$, the average maturation proportion of BMDCs (29.7%) reached a peak. Exceeded doses tended to display a slow increase, indicating that too

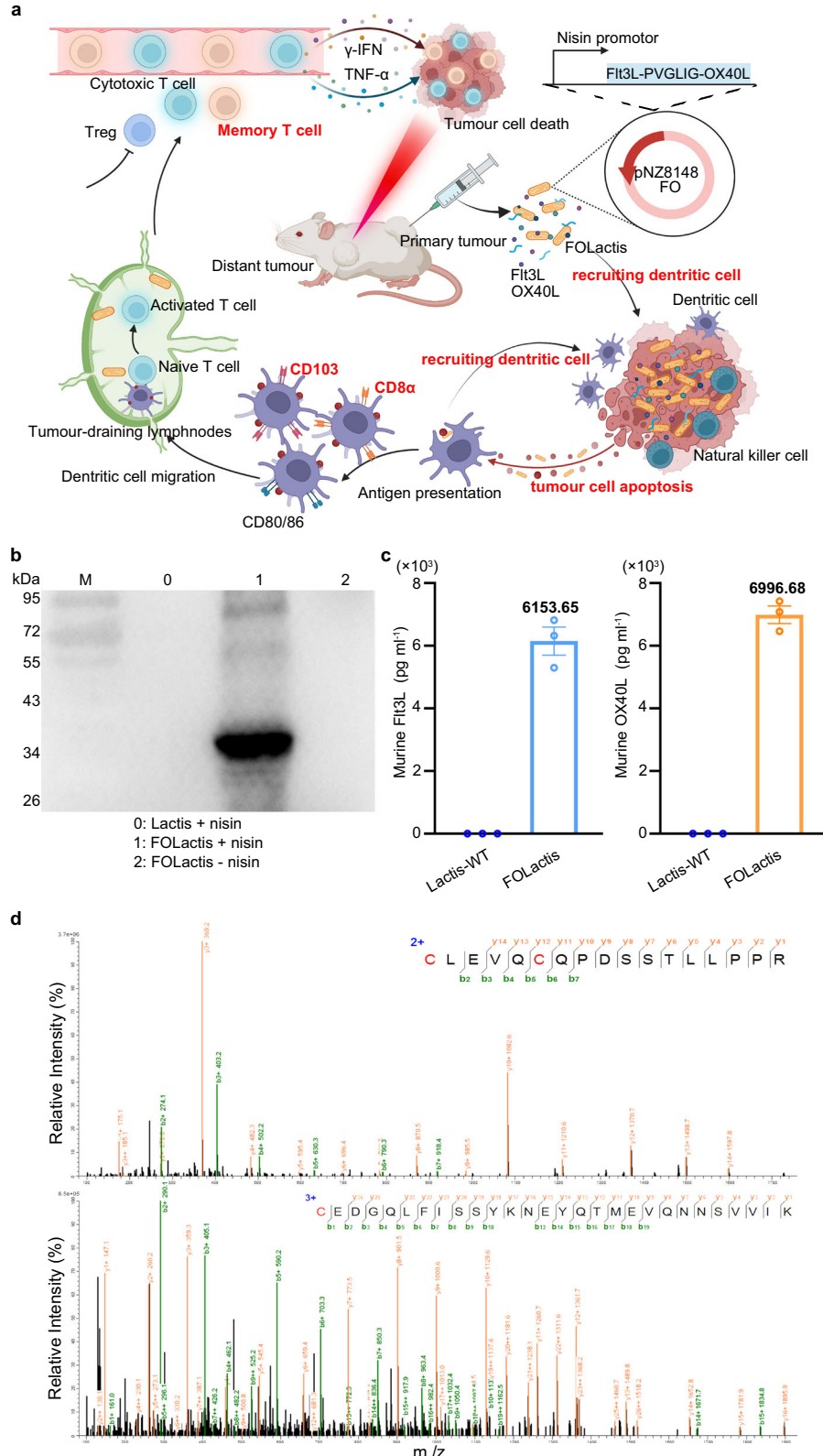

**Fig. 1 | Construction of engineered *Lactococcus lactis* delivering Flt3L-OX40L fusion protein (FOLactis). a** Schematic diagram of intratumourally injecting FOLactis to enhance tumor immunotherapy by reprogramming the immune microenvironment of local tumor and tumor-draining lymphnodes. Created with BioRender.com. **b** Western blotting analysis of the induced or non-induced engineered *Lactococcus lactis* (One representative data was shown from three independently repeated experiments). Nisin is an inducer of protein expression. M: molecular mass marker; Lane 0–2: the whole bacteria lysates (bacteria) of wild-type *Lactococcus lactis* (Lactis) induced by nisin, FOLactis induced by nisin and non-induced FOLactis, respectively. **c** Bacteria (10⁹ CFU) were collected and the pellets were sonicated. The amounts of the target protein in the bacterial lysates of FOLactis were assessed by ELISA (*n* = 3, biologically independent samples). The error bars represented mean ± s.e.m. **d** Fingerprints of the two best matched peptides. Source data are provided as a Source Data file.

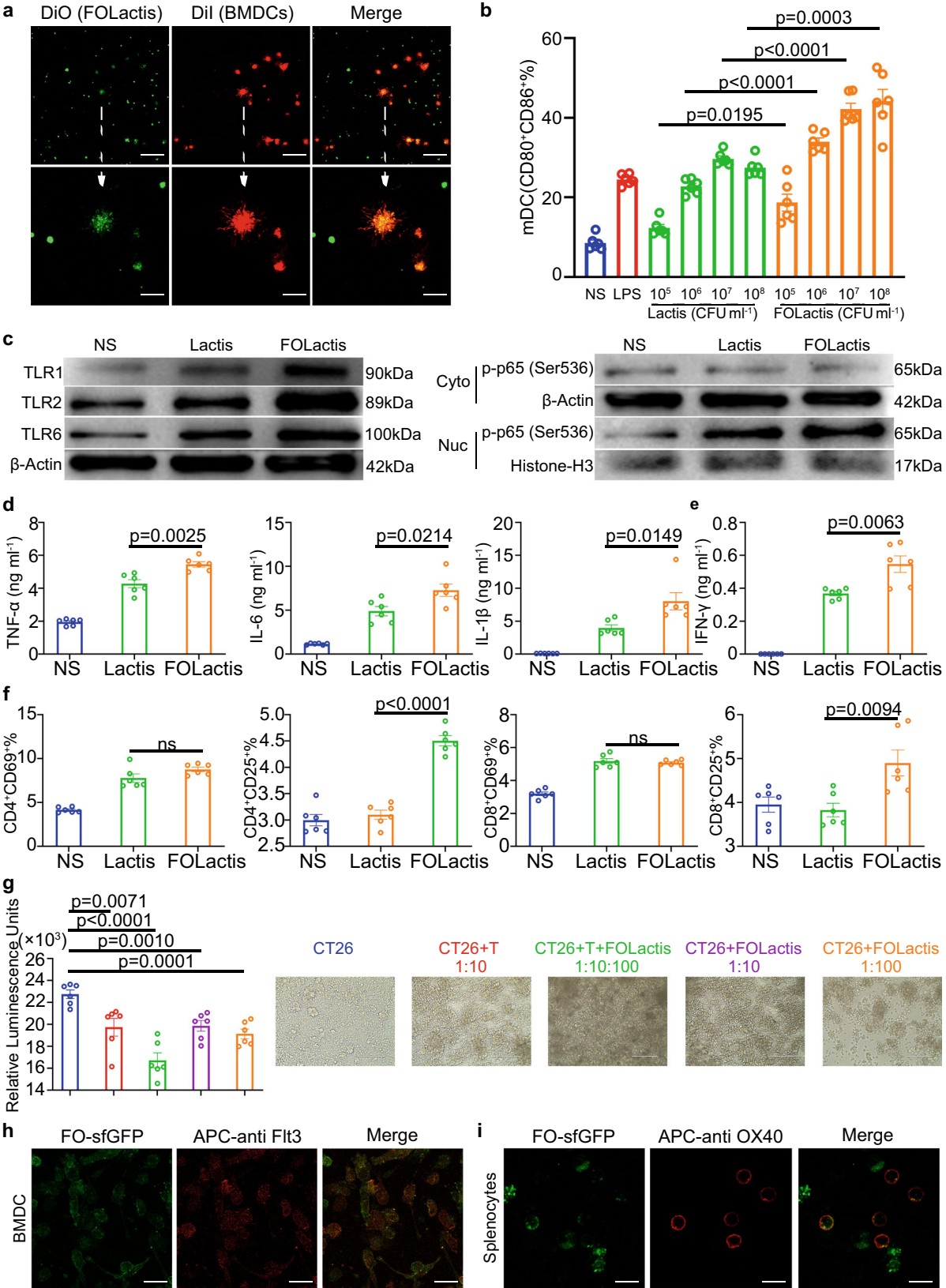

high a concentration might cause side effects on the cells (Fig. 2b). Considering the efficacy and safety, we finally chose 10⁷ CFU ml⁻¹ Lactis and FOLactis in the following study (Fig. 2c–f). BMDCs tend to display a number of pattern recognition receptors, especially TLRs. These receptors recognize LAB-specific pathogen-associated molecular patterns (PAMPs), facilitating the activation of diverse cascades of intracellular signaling, such as NF-κB transcription factor, with the resultant synthesis of proinflammatory cytokines and chemokines[22,38]. We observed FOLactis induced a higher expression of TLR1, TLR2 and TLR6 in BMDCs than that in the NS group, which further activated NF-κB signaling pathway (Fig. 2c). Meanwhile, the secretion of TNF-α, IL-6 and IL-1β significantly increased in the FOLactis group compared to

**Fig. 2 | FOLactis efficiently enhanced the activation of DCs and T cells, and induced tumor cell death in vitro. a** Colocalization analysis of FOLactis (DiO; green) in bone marrow-derived DCs (BMDCs) (DiI; red) by confocal microscopy (two-hour incubation) (One representative data was shown from three independently repeated experiments). White scale bars: Up, 50 μm; Down, 10 μm. **b** The percentage of mature DCs (mDCs, CD11c⁺CD80⁺CD86⁺) after co-incubation with Lactis or FOLactis in vitro for 24 h ($n = 6$, biologically independent samples). **c** Western blot analysis of several membrane TLR proteins and NF-κB pathway in BMDCs. The gels were loaded with equal amounts of the proteins (10 μg) (One representative data was shown from three independently repeated experiments). **d** Pro inflammatory cytokine concentrations in BMDC supernatants ($n = 6$, biologically independent samples). **e** Assessment of IFN-γ in coculture supernatants after T cells from the spleen of BALB/c mouse stimulated by Lactis or FOLactis for 24 hours in vitro ($n = 6$, biologically independent samples). **f** The quantification of CD69 and CD25 expression on CD8⁺ and CD4⁺ T-cell subsets in the T cells ($n = 6$,

biologically independent samples). **g** Luminescence was detected by the addition of D-luciferin to CT26-GFP-Luc cells after incubated with T cells and/or FOLactis for 24 h ($n = 6$, biologically independent samples). Cells were examined under the microscope EVOS FL Auto Cell Imaging System (Invitrogen) and photographed. White scale bars, 200 μm. **h, i** The Flt3L-OX40L-sfGFP fusion protein (FO-sfGFP) was extracted from lysates of FOLactis-sfGFP. Confocal laser scanning microscopy images of mouse BMDCs treated with FO-sfGFP and allophycocyanin (APC)-anti mouse Flt3 for two hours. Confocal laser scanning microscopy images of mouse splenocytes treated with FO-sfGFP and APC anti-mouse OX40 for two hours (One representative data was shown from three independently repeated experiments). FO-sfGFP, green; APC-anti mouse Flt3 (APC anti-mouse OX40), red. White scale bars, Left, 50 μm; Right, 10 μm. For **b, d–g**, data were mean ± s.e.m. statistical significance was determined by two-tailed unpaired Student's t-tests. ns represented $p > 0.05$. Source data are provided as a Source Data file.

the other groups, which played crucial roles in the initiation and stimulation of innate immune response (Fig. 2d). Moreover, when we incubated BMDCs with different concentrations of FO and the crude lysates from Lactis or FOLactis in vitro for 24 h, markedly enhanced expression of CD80 and CD86 was observed on mature DCs. When the concentration of FO or FOLactis reached 50 μg ml⁻¹ or 10⁷ CFU ml⁻¹, the average maturation proportion of BMDCs was 56.3% and 49.2%, respectively (Supplementary Fig. 5a, b). These results indicated that both FOLactis and their debris could act as immune stimulatory agents.

Splenocytes of the BALB/c mice were used as a source of DCs and T cells to induce activated or memory T cells. After treated with live Lactis or FOLactis for 24 h in vitro, both Lactis and FOLactis induced a higher expression of CD69 (early activation marker) than that in the NS group. We also observed a higher expression of CD25 (late activation marker) in the FOLactis group than that in the Lactis group, leading to the robust generation of IFN-γ (Fig. 2e, f). In the presence of autologous tumor cell membranes (TM), CD4⁺ and CD8⁺ T cells activated by FO or the crude lysates from FOLactis significantly had at least about a three-fold increase in surface expressions of CD69 and CD25 compared with the TM group (Supplementary Fig. 5d–g and Supplementary Fig. 6b–e). Consistent with the activation of T cells, FOLactis promoted the differentiation of naïve T cells into the effector memory T cells (T$_{EM}$) by more than two-fold compared to the TM group (Supplementary Fig. 5c and Supplementary Fig. 6a).

To determine whether FOLactis could kill cancer cells by direct cell lysis, we observed the morphological changes of cancer cells after treatment with FOLactis and/or splenocytes in vitro for 24 h. Notably, these cancer cells produced large quantities of balloon-shaped vesicles and the luciferase activity decreased (Fig. 2g and Supplementary Fig. 7a). Cell death induction was further confirmed by increased staining with Propidium Iodide (PI), a cell impermeable dye that stains cells that have lost membrane integrity (Supplementary Fig. 7b).

**Temporal colonization and biodistribution of FOLactis**
To determine the optimal concentration of FOLactis for i.t. injection, we implanted BALB/c mice with CT26 tumor cells subcutaneously (s.c.). When the tumors were palpable (~100 mm³), previously induced FOLactis were injected (i.t.) at increasing colony-forming unit (CFU) concentrations dissolved in 100 μl NS. The tumor growth of the 10⁹ CFU group and 10¹⁰ CFU group was greatly inhibited compared to the other three groups of NS, 10⁷ CFU, and 10⁸ CFU (Fig. 3e, g). Consistently, the survival of mice in the 10⁹ CFU group and 10¹⁰ CFU group was significantly longer than that of any other groups (Fig. 3h). Because the mouse body weights showed a rapid drop in the 10¹⁰ CFU group (Fig. 3f), we finally treated mice with 10⁹ CFU FOLactis.

Similar to the construction of FOLactis, we designed the engineered Lactis expressing Flt3L (FLactis) or OX40L (OLactis) (Supplementary Fig. 8a, b). The successful expression of the protein was confirmed by WB and ELISA (Supplementary Fig. 8c). It was found

that the mice in the FOLactis group exhibited a stronger inhibition of tumor growth compared to the mice in other groups, with a marked extension of animal survival over the other groups (Supplementary Fig. 8d, e).

To confirm the bacterial motility in vivo, we injected DiR-labeled bacteria intratumorally once to visualize the distribution of FOLactis. Strikingly, we found that all injected bacteria were mainly trapped in the tumor core, which peaked at 72 h and then declined until 360 hours after treatment (Fig. 3a–c). Tumors and TDLNs were resected at different time points (6, 24, 72, 168, 360 h) and measured ex vivo. While the fluorescence was weaker in TDLNs than in tumors, the signal was detected early at six hours and lasted for at least 360 h (Fig. 3a, b, d), which might provide an excellent opportunity for immune activation in TDLNs. In addition, there was almost no fluorescent signal in other organs, including the heart, liver, spleen, lung and kidney (Supplementary Fig. 9), indicating the locoregional administration of FOLactis had a good biological safety. The similar results was shown when we cultured and enumerated FOLactis on solid GM17 agar plates (Supplementary Fig. 10a, b).

**Injection (i.t.) of FOLactis exerted an ISV effect in a murine colon cancer model**
Since we have demonstrated the tumor growth inhibition efficacy of i.t. injection of 10⁹ CFU FOLactis, we used CT26 colon cancer cells to establish a mouse model with tumors at two different sites to observe its ISV effect, that was, the systemic immunity elicited by injection (i.t.) of FOLactis in the primary tumor also inhibited the growth of the untreated one. The concentration of FO was finally 50 μg in vivo considering the antitumour efficacy (Supplementary Fig. 11a–c). As shown in Fig. 4a, the same number of CT26 cells were inoculated on the left and right sides of the abdomen of BALB/c mice at a three-day interval. Seven days after the first inoculation, when the primary tumor volume reached about 100 mm³, they were randomly divided into five groups for treatments: NS, Lactis (10⁹ CFU Lactis), FO (50 μg FO), Lactis + FO (10⁹ CFU Lactis + 50 μg FO), FOLactis (10⁹ CFU FOLactis). All the drugs were dissolved in 100 μl NS before injection into the primary tumor. The treatment was repeated twice at a two-day interval after the first injection. Then we monitored the tumor growth at the treated and untreated sites with calipers.

The tumor growth curves at treated sites in the Lactis + FO and FOLactis groups were significantly suppressed compared with the other groups, especially in the FOLactis group. In the FOLactis group, all the tumors were below 200 mm³, and 50% (4/8) of mice achieved complete regression (CR). These results suggested that both FO and Lactis exerted antitumour effects, and the engineered FOLactis was superior to a simple blend of FO and Lactis (Fig. 4b, d). Surprisingly, there was also a significant difference in untreated tumors between the FOLactis and the other four groups. Similar to the treated tumors, the FOLactis group displayed the best tumor control with a 37.5% (3/8) CR

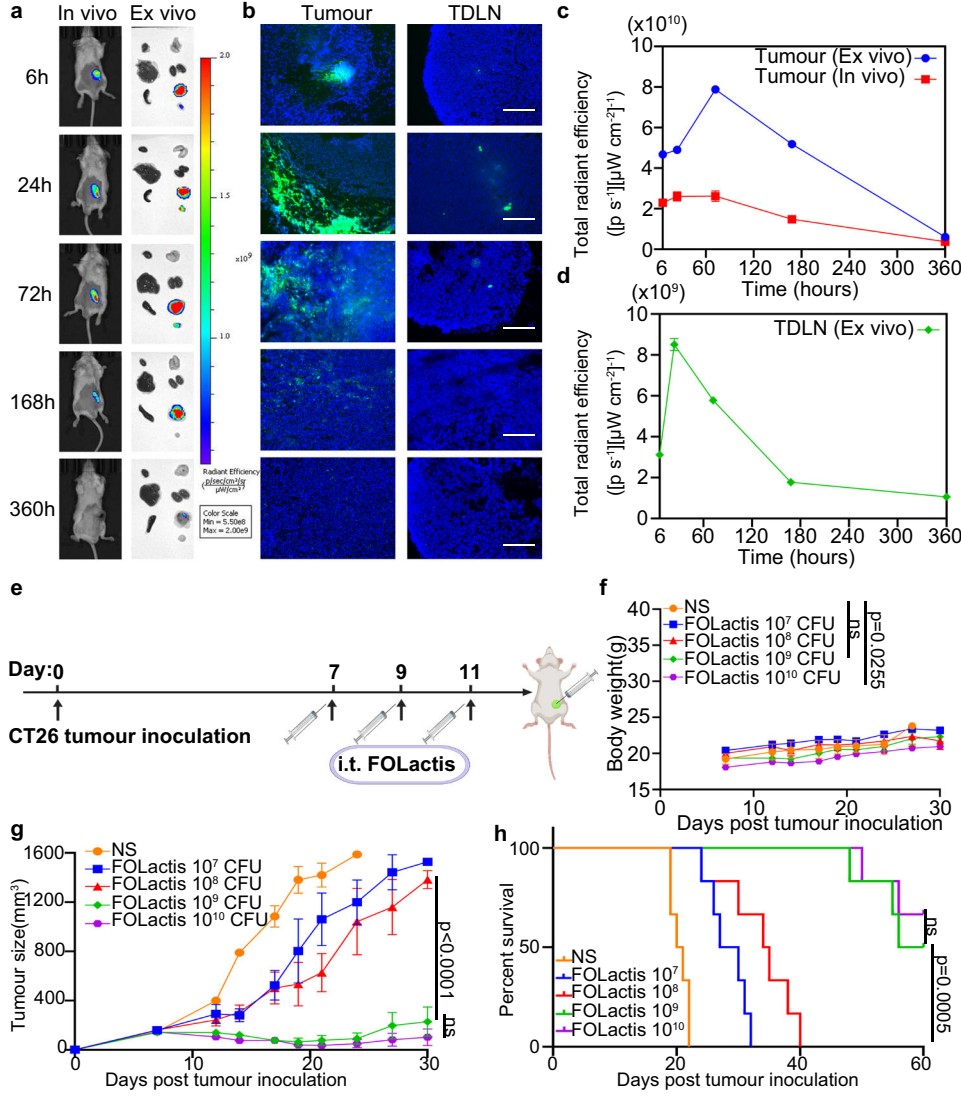

**Fig. 3 | Temporal colonization and biodistribution of FOLactis. a** Left, Representative in vivo near-infrared imaging of tumor-bearing mice at 6, 24, 72, 168, 360 h after intratumoural injection of $10^9$ CFU FOLactis ($n = 3$). Right, Representative ex vivo images of the hearts, livers, spleens, lungs, kidneys, tumors and TDLNs collected at 6, 24, 72, 168, 360 h after injection ($n = 3$). **b** Frozen sections of tumors and TDLNs in CT26 mouse colon tumor model at 6, 24, 72, 168, 360 h after intratumoural injection of $10^9$ CFU FOLactis ($n = 3$). FOLactis were labeled with DiO (green); nucleus, blue. White scale bars, 200 μm. **c** Total radiant efficiency of FOLactis signal in tumors in vivo or ex vivo over time ($n = 3$, biologically independent samples). The error bars represented mean ± s.e.m. **d** Total radiant efficiency of FOLactis signal in TDLNs ex vivo over time ($n = 3$, biologically independent samples). The error bars represented mean ± s.e.m. **e** Schematic diagram of the treatment in male CT26 tumor-bearing mice. BALB/c mice were implanted with CT26 cells ($5 \times 10^5$) on the left lower sides of the abdomen on day 0, and received

treatments on days 7, 9 and 11. Created with BioRender.com. **f** Average weight of different groups for 30 days ($n = 4$). The error bars represented mean ± s.e.m. *p*-values were calculated by two-tailed unpaired Student's t-tests. ns represented $p > 0.05$. **g** Average tumor-growth curves of BALB/c mice bearing CT26 colon tumor with different treatments as indicated ($n = 4$). The mice were administered with NS, $10^7$ CFU FOLactis, $10^8$ CFU FOLactis, $10^9$ CFU FOLactis or $10^{10}$ CFU FOLactis intratumourally on days 7, 9 and 11, which were dissolved in normal saline to a final volume of 100 μl per dose. The tumor size was measured every 2–3 days from the first administration day. The error bars represented mean ± s.e.m. *p*-values were calculated by two-way ANOVA and Tukey post-test and correction. ns represented $p > 0.05$. **h** Survival curves of BALB/c mice in different groups for 60 days ($n = 6$). *p*-values were calculated by log-rank (Mantel-Cox) test. ns represented $p > 0.05$. Source data are provided as a Source Data file.

rate at the untreated sites (Fig. 4c, e). Simultaneous and effective suppression of tumors at both sites resulted in prolonged survival of mice in the FOLactis group. 70% (7/10) of mice in the FOLactis group survived for 60 days, while only 30% (3/10) in the Lactis + FO group (Fig. 4f). The body weight change in all groups had a similar changing pattern during the treatment (Fig. 4g).

### Injection (i.t.) of FOLactis induced immune response in the tumor and TDLN

In the previous tumor suppression experiments, we found that intratumourally delivered FOLactis exerted an ISV effect, inhibiting

untreated tumors. To assess the immune response elicited by the ISV effect of FOLactis, we repeated the tumor suppression experiment. Subsequently, we removed intratumourally treated tumors and TDLNs of mice in the five groups after two days from the last treatment to detect the changes in CD103$^+$ DCs and CD8$^+$ DCs. Immunofluorescence images showed that there were almost no CD103$^+$ DCs in the tumor of WT Lactis. In contrast, the engineered FOLactis significantly increased the infiltration of CD103$^+$ DCs, indicating that the production of FO was a critical factor in improving cross-presentation ability of DCs (Fig. 5a and Supplementary Fig. 14a). We further confirmed this result by flow cytometry assays. In treated tumors, the proportion of CD103$^+$ DCs in

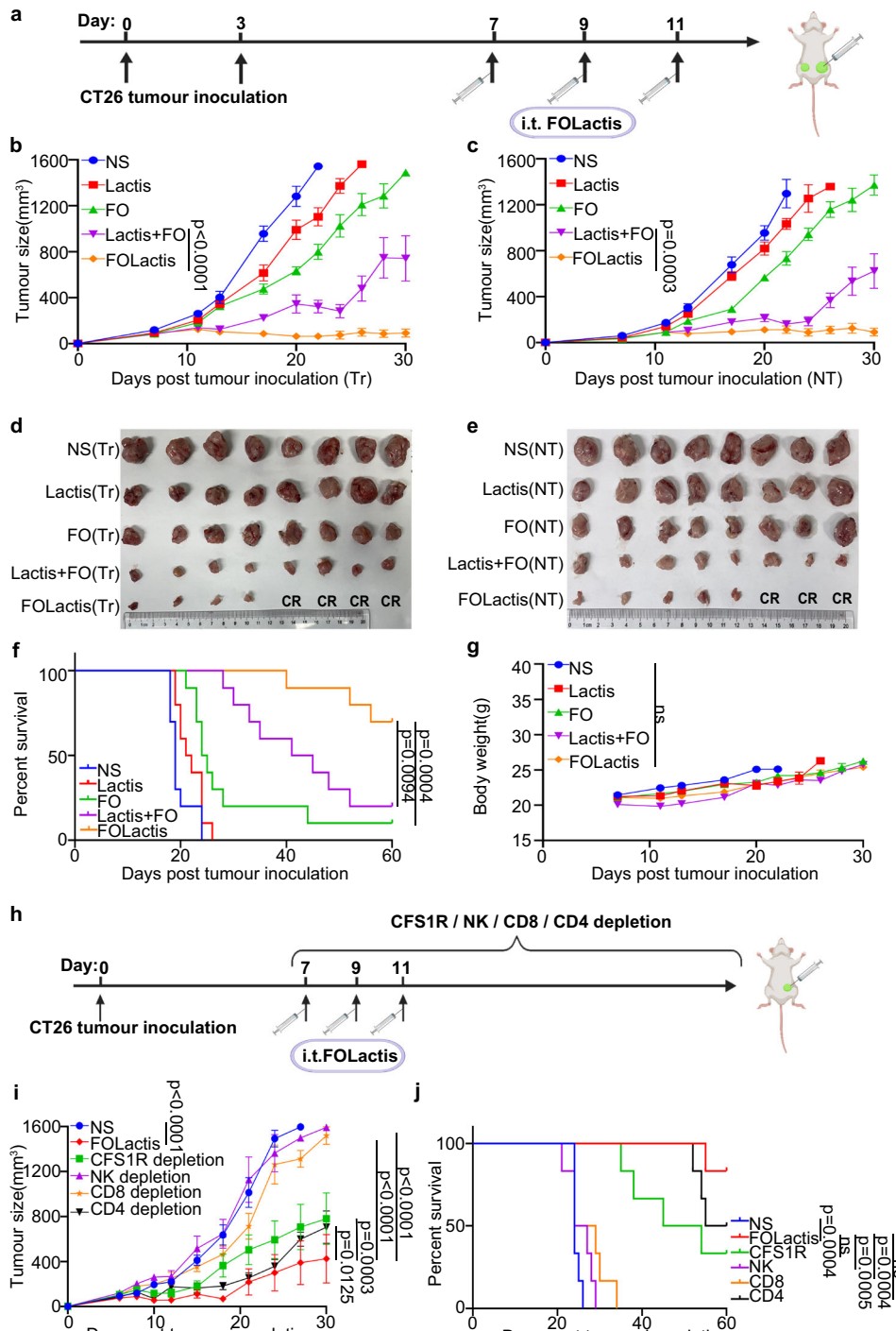

**Fig. 4 | ISV with FOLactis potently inhibited the treated and untreated tumor growth simultaneously in a model of Colon Cancer. a** Schematic diagram of the treatment in male CT26 tumor-bearing mice. Created with BioRender.com. Average tumor-growth curves (**b**: Treated tumors (Tr). **c**: Nontreated tumors (NT).) of BALB/c mice bearing CT26 colon tumor with different treatments as indicated ($n = 8$). The mice were administered with NS, $10^9$ CFU Lactis, 50 μg FO, $10^9$ CFU Lactis + 50 μg FO, and $10^9$ CFU FOLactis dissolved in NS to a final volume of 100 μl per dose intratumourally on days 7, 9, and 11. The tumor size was measured every 2-3 days from the first day of administration. **d**, **e** Photos of tumors harvested from mice in all groups on day 26 after tumor inoculation ($n = 8$). CR: complete response. **f** Survival curves of BALB/c mice in different groups for 60 days ($n = 10$). P-values were calculated by log-rank (Mantel−Cox) test. ns represented $p > 0.05$. **g** Average weight of different groups for 30 days ($n = 8$). The error bars represented mean ± s.e.m. p-values were calculated by two-tailed unpaired Student's t-tests. ns

represented $p > 0.05$. **h** Scheme showing the design of the animal experiment. BALB/c mice were subcutaneously inoculated with CT26 colon adenocarcinoma cells ($5 \times 10^5$ per mouse) and randomized into six groups ($n = 5$ mice per group) with their tumors treated with: (i) NS, (ii) $10^9$ CFU FOLactis, (iii) $10^9$ CFU FOLactis and depleting antibody CFS1R, (iv) $10^9$ CFU FOLactis and depleting antibody NK, (v) $10^9$ CFU FOLactis and depleting antibody CD8, (vi) $10^9$ CFU FOLactis and depleting antibody CD4. Created with BioRender.com. Tumor growth curves (**i**) and survival data (**j**) of CT26 tumor-bearing mice after different treatments indicated. For the experiments in **b**, **c**, **i**, data was the mean ± s.e.m. p-values were determined by two-way ANOVA with Tukey's multiple comparisons test. ns represented $p > 0.05$. Differences in survival were determined by using the Kaplan-Meier method, and the p value was calculated via the log-rank (Mantel−Cox) test. Source data are provided as a Source Data file.

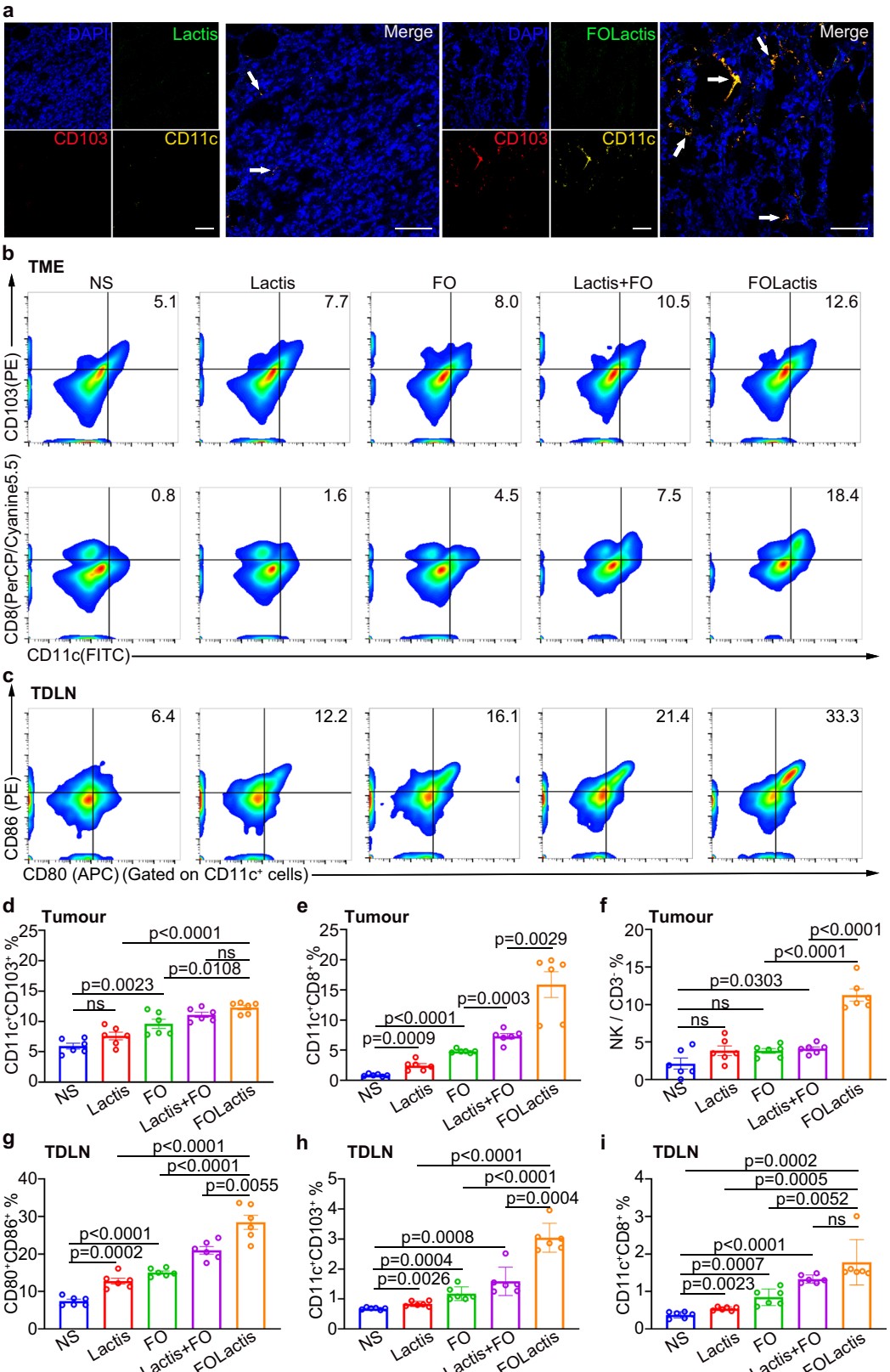

**Fig. 5 | ISV with FOLactis reprogrammed the immune microenvironment in the local tumor and TDLN. a** Ex vivo immunofluorescence staining of CD11c⁺ CD103⁺ DCs in the treated tumor 24 h after intratumoural injection of Lactis or FOLactis (1 × 10⁹ CFU per mouse). Lactis or FOLactis, DCs, and tumor nuclei were labeled with DiO (green), anti-CD11c-PE antibody (yellow), anti-CD103-APC antibody (red), and DAPI (blue), respectively. Scale bars, 50 μm (*n* = 3). **b, c** CT26 tumor-bearing mice in different groups were sacrificed two days after the last treatment, and the proportions of immune cells were analyzed by flow cytometry in the local tumor and TDLN

(*n* = 6). Shown are representative flow plots of CD11c⁺CD103⁺ or CD11c⁺CD8⁺ DCs subpopulation (**b**) and CD80⁺CD86⁺ mDCs gated on CD11c⁺ DCs (**c**). Flow cytometric analysis of CD11c⁺CD103⁺ DCs (**d**), CD11c⁺CD8⁺ DCs (**e**) and natural killer cells (**f**, gate: CD3⁻ T cell) in the tumor microenvironment (TME) (*n* = 6). Flow cytometric analysis of CD80⁺CD86⁺ mDCs (**g**, gate: CD11c⁺ DCs), CD11c⁺CD103⁺ DCs (**h**) and CD11c⁺CD8⁺ DCs (**i**) in the TDLN (*n* = 6). For the experiments in **d–i**, data were the mean ± s.e.m. statistical significance was determined by analysis of two-tailed unpaired Student's t-tests. ns represented *p* > 0.05. Source data are provided as a Source Data file.

the FOLactis group was highest and had a nearly two-fold increase compared to the NS group, and the ratio of CD8$^+$ DCs showed even a 20-fold increase in the FOLactis group (Fig. 5b, d, e). TDLNs are crucial for tumor antigen-loaded DCs to activate naïve T cells. As shown in Fig. 5c, g–i, the ISV effect caused by FOLactis induced a large number of mature DCs in TDLNs. The upregulation of CD80$^+$CD86$^+$ indicated a more mature stage of DCs, among which the functional CD103$^+$ DCs and CD8$^+$ DCs were even increased by two-fold compared to the Lactis + FO group. We further excluded the possibility of bacterial trafficking by assessing the biodistribution of FOLactis at different time points after unilateral intratumoural injection. It was found that bacterial growth remained restricted to treated tumors and almost no bacteria could be cultured from untreated tumors or the other major organs of treated mice (Supplementary Fig. 10a, b). Additionally, the proportion of T$_{EM}$ (CD3$^+$CD8$^+$CD44$^+$CD62L$^-$) subset and IFN-γ$^+$ CD8$^+$ T cells increased at the untreated site 7 days later in the FOLactis group compared to the NS group (Supplementary Fig. 12j–l).

In addition to the infiltration of various DCs within the tumors and TDLNs, we also investigated NK cells and tumor-associated macrophages (TAMs), which played essential roles in innate immunity. There was a significantly higher proportion of NK cells in the Lactis+FO group and FOLactis group compared to the NS group, suggesting that Lactis combined with FO contributed to the increase of NK cells. (Fig. 5f). Both Lactis and FO polarized TAMs from immunosuppressive type 2 macrophages (M2, CD11b$^+$F4/80$^+$CD206$^+$) to immune promoting type 1 macrophages (M1, CD11b$^+$F4/80$^+$CD86$^+$). Compared to the NS group (3.5%), the proportion of M1 in the Lactis group and FO group increased to 6.2 and 7.4%. Engineered FOLactis maximized their synergy, reaching 16.8% in M1, higher than the Lactis + FO group (10.2%) (Supplementary Fig. 12a, b).

The improvement of DCs, NKs, and TAMs finally promoted the infiltration of effective T cells in TDLNs and tumors. The CD8-to-Treg ratio in TDLNs went to 17.3%, almost three-fold than that in the NS group (Supplementary Fig. 12g–i). Eventually, CD8$^+$ TILs significantly increased from 3.7% in NS to 13.8% in the FOLactis group and CD4$^+$ TILs also had a slight increase (Supplementary Fig. 12c, d).

Besides, we also detected cellular pro-inflammatory cytokines in treated tumors, including interferon-γ (IFN-γ), TNF-α, and IL-6. The secretion of the three cytokines in the FOLactis group was significantly higher than that in the other four groups (Supplementary Fig. 13). Our results illustrated that FOLactis reprogrammed the microenvironment to be more immunogenic in the local tumor and TDLNs.

## Innate and adaptive immunity were both required for tumor rejection after ISV with FOLactis

Long-term tumor regression tends to require both innate and adaptive immune responses. NK cells and macrophages are essential for innate immunity, whereas CD4$^+$ T cells and CD8$^+$ T cells play vital roles in adaptive immune response[39]. To illuminate the underlying mechanisms of the enhanced tumor rejection in response to i.t. injection of FOLactis, we performed the RNA sequencing (RNA-seq)-based transcriptome analyses of the treated tumor. There were 1745 differently expressed genes (DEGs), 1489 significantly up-regulated and 256 down-regulated genes (Fig. 6a). The Gene Ontology (GO) and Kyoto Encyclopedia of Genes and Genomes (KEGG) functional enrichment analyses identified significantly enriched pathways in DEGs, including interleukin-6 (IL-6) secretion, interferon-gamma-mediated signaling pathway, TLR signaling pathway, NK cell mediated cytotoxicity, NOD-like receptor signaling pathway and positive regulation of activated T cell proliferation (Fig. 6b, d, e), which were related with the activation of both innate and adaptive immune responses. As shown in the gene network, many up-regulated genes, especially IL-6, IFN-γ, TLR2, CXCL10, NOD2, etc., were associated with the activation of T cells and NK cells (Fig. 6c).

To validate the DEG results in vivo, we randomly divided the mice inoculated with CT26 cells into six different groups with/without depleting antibodies against cell surface markers. The depleting antibodies were administered intraperitoneally (i.p.) one day before treatments. Mice were also injected (i.t.) with 10$^9$ CFU FOLactis three times on days 7, 9, and 11, except for those in the NS group (Fig. 4h). Notably, the depletion of NK cells or CD8$^+$ T cells led to a tumor growth similar to the NS group, whereas FOLactis significantly inhibited the tumor growth ($p < 0.0001$). Accordingly, the overall survival rates of mice in the two depletion groups decreased remarkably ($p = 0.0005$, $p = 0.0004$). In contrast, for mice treated with macrophages or CD4$^+$ T cells depleting antibodies and FOLactis, the observed antitumour effects, although strong enough to delay the growth of tumors, were far from comparable to that achieved in the FOLactis group (Fig. 4i, j). Overall, ISV with FOLactis could induce tumor regression mainly dependent on CD8$^+$ T and NK cells, with other immune cell subsets, such as macrophages and CD4$^+$ T cells, playing lesser roles in tumor rejection.

## ISV with FOLactis led to long-term immunological memory in a syngeneic tumor model

To further confirm ISV with FOLactis could provide long-term protection over 60 days, we collected splenocytes for memory T cells analysis after i.t. injection of FOLactis in a CT26 tumor model. We found that the percentage of T$_{EM}$ (CD3$^+$CD8$^+$CD44$^+$CD62L$^-$) was much higher (56.8%) in the FOLactis group than that in the NS (27.9%) or Lactis groups (35.5%). T$_{EM}$ is the leading force for the production of IFN-γ and TNF-α[40]. The T$_{EM}$ of the FOLactis group was not significantly higher than that in the FO or FO + Lactis group, but the survival was still most favorable in the FOLactis group. We think there were two important factors contributed: One was that Lactis promoted the expression of OX40L on T cells, thus exhibiting a synergistic effect with OX40; the other was that the FO protein was linked to Lactis in the engineered FOLactis, which increased the half-life of FO to exert their antitumour effect fully, and this explained why FOLactis was more effective than FO + Lactis (Fig. 7d–f and Supplementary Fig. 12e, f).

Furthermore, we isolated lymphocytes from the mice spleen in the NS and FOLactis groups and cocultured them with CT26 tumor cells for 12 h. The number of PI$^+$ cancer cells showed an almost three-fold upregulation when incubated with lymphocytes of the FOLactis group at an effector-to-target ratio (E: T) of 5:1, indicating that FOLactis administration armed lymphocytes with more potent cytotoxic activity to kill cancer cells directly (Fig. 7g). Moreover, FOLactis-treated and cured mice were rechallenged with the CT26 tumors after two months (Fig. 7k). Notably, while the tumor of untreated mice grew rapidly, the FOLactis-treated mice showed a tumor inhibition rate of 100%, and all of them could survive for another 60 days (Fig. 7l–n).

## ISV with FOLactis upregulated checkpoint molecules in TME and synergized with anti-PD1 antibody

Despite tumor regression and immune activation after treatment with FOLactis, some tumors were still growing at a slow rate or recurred within 30 days after the ISV (Fig. 4b, c), suggesting an immune evasion of tumor cells. In the previous tumor suppression experiments, we observed an elevated expression of PD1 on CD8$^+$ TILs in FOLactis-treated group compared to the NS group within seven days after the first injection (Fig. 7a–c and Supplementary Fig. 14b). To evaluate whether PD1 blockade would improve antitumour immune responses, FOLactis-treated mice received anti-PD1 monoclonal antibody (i.p.) three times, as illustrated in Fig. 7h. In combination with anti-PD1, FOLactis further suppressed the growth of treated and untreated tumors in a 40-day observation after the tumor inoculation (Fig. 7i). Consistently, while all the mice in the NS and anti-PD1 groups died within 26 days, the combination therapy of FOLactis + anti-PD1 increased the percent survival from 60% in FOLactis to 100% (Fig. 7j). These results indicated the potential for the combination therapy of FOLactis and immune checkpoint inhibitors in clinical application.

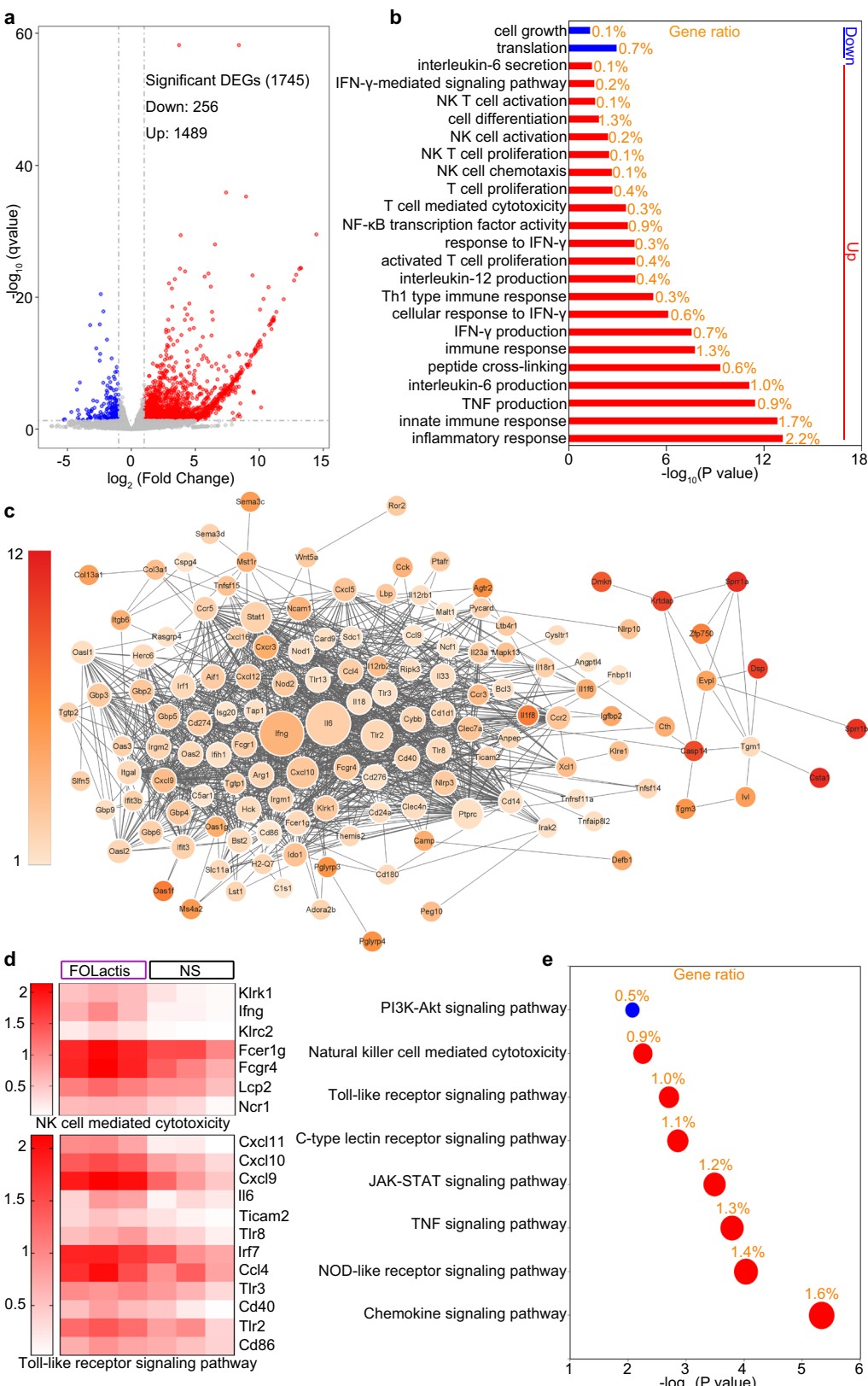

### ISV with FOLactis initiated antigen-specific therapeutic effects and could inhibit multiple immunologically "cold" tumor types

In addition to the CT26 tumor models, which tended to be comparatively responsive to immunotherapy[41], we also demonstrated the universal applicability of this ISV in two 'cold' tumor models, the B16F10-OVA melanoma and 4T1 breast tumor mouse models.

In alignment with the results in CT26 tumor-bearing BALB/c mice, the ISV with FOLactis exhibited the best efficacy in tumor suppression and survival compared to other groups (Fig. 8a–c, f–h). Furthermore, this was confirmed with the study of tumor growth and metastasis in the orthotopic 4T1 tumor models. The results showed that FOLactis could significantly reduce lung metastasis and

**Fig. 6 | RNA-seq analysis of tumors after ISV with FOLactis. a**, Volcano map of differentially expressed genes (DEGs) between the NS and FOLactis groups ($n = 3$). The $x$-axis was the log2 scale of the fold change of gene expression. Negative values indicated downregulation; positive values indicated upregulation. The $y$-axis was the minus log10 scale of $q$ values (the adjusted $p$ values), indicating the significant expression difference level. The red dots represented significantly upregulated genes with at least two-fold change, while the blue dots represented significantly downregulated genes with at least two-fold change. **b** Gene Ontology (GO) analysis of up- and downregulated genes. Gene ratio (shown in orange) and $-\log_{10}$(P value) of all GO terms were shown. **c** GO network analysis of significantly upregulated genes in tumors from the FOLactis-treated mice, as compared with the NS-treated mice. Color scales represented the values of $\log_2$-transformed fold changes. **d** Differential genes related to Natural killer cell mediated cytotoxicity or Toll-like receptor signaling pathway were analyzed by RNA sequencing. The high-expression genes and low-expression genes were clustered by $\log_{10}$ (FPKM+1). The color from white to red indicated that the gene expression from low to high. **e** KEGG enrichment analysis. Source data are provided as a Source Data file.

increased the survival rate of tumor-bearing mice (Supplementary Fig. 15a–e).

Next, OVA-expressing B16F10 melanoma mouse models were used to explore the initiation of tumor-specific immune responses. Lymphocytes were isolated from the treated tumor two days after the last treatment to examine the generation of model tumor antigen (OVA)-specific CTLs by staining with OVA/H-2K$^b$ tetramer. Importantly, the average frequency of OVA-tetramer$^+$ CD8$^+$ CTLs in the FOLactis group elevated from 0.8 to 1.7% compared to the levels in the NS group ($p = 0.0081$). FOLactis also generated almost two-fold higher antigen-specific CD8$^+$ T cell responses than Lactis alone or FO alone (Fig. 8e).

We further harvested the treated 4T1 breast tumor to evaluate the therapeutic efficacy of this local immunotherapy by immunohistochemical analysis. The number of CD8$^+$ and PD1$^+$ T cells significantly increased in TME seven days after i.t. injection of FOLactis (Fig. 8i, j and Supplementary Fig. 14c). These results demonstrated the power of the ISV to successfully convert the 'cold' tumors to 'hot' tumors, thus exerting a robust antitumour effect.

### Biosafety assessment of ISV with FOLactis

Injection (i.t.) of bacteria may pose potential risks, so it is necessary to verify the safety of ISV with FOLactis carefully. When it was applied to treat different mouse tumor models, mice in all groups did not experience abnormal weight fluctuations (Figs. 4g, 8d). H&E-stained images of the main organs (heart, liver, spleen, lung, kidney) taken out on the 7th day after the last treatment showed no apparent damage in all groups (Supplementary Fig. 16a). Besides, we performed hematological examinations, including serum biochemistry assays. There was no significant difference in the serum biochemistry indexes of the mice in each group, and all of them were within the normal range (Supplementary Fig. 16b).

During immunotherapy, life-threatening side effects, such as a cytokine storm, may occur. Therefore, we measured the concentrations of inflammatory cytokines and chemokines in the serum two days and seven days after the last treatment, evaluating the response of systemic inflammation stimulated by the FOLactis. Overall, the serum concentrations of IFNg, IL-2, IL-13, IL-4, IL-10, IL-5, IL-6, and TNF-α were generally similar among all the groups without a statistical difference, indicating that ISV with FOLactis had a good biological safety (Supplementary Fig. 17 and Supplementary Fig. 18).

### Discussion

Here, we designed a bifunctional engineered *Lactococcus lactis*-based in-situ vaccine to elicit systemic tumor regressions by rejuvenating DCs and cytotoxic T cells. We focused on safety, practicability and effectiveness in the species and strain of bacteria selection. Compared to some attenuated strains, which might risk toxicity recurrence, LAB has a long history as raw materials for fermented food products and is generally recognized as safe (GRAS). Some of the LAB strains are the components of the gut's ecosystem, which are associated with the enhanced efficacy of immunotherapy, so they might be promising therapeutic strategies for clinical application. The most widely used strain among LAB is *Lactococcus lactis* because of its relatively mature and straightforward genetic toolbox. IL-10 secreting *Lactococcus lactis* was the first genetically engineered therapeutic bacterium used in

human trials, in which most of the patients with Crohn's disease showed improved clinical parameters without severe adverse events[42]. A big step forward in the engineering of *Lactococcus lactis* is the invention of Nisin Controlled Gene Expression System (NICE®)[43], based on which protein production greatly increased. After many considerations and tests of the above three key points, we chose the NICE® system to deliver antitumour agents Flt3L and OX40L.

Immunosuppressive TME is a big hurdle that may limit the antitumour effect of cancer treatment[44,45], where DCs and T cells play pivotal roles. Priming and activation of CD8$^+$ antitumour T cell responses against solid tumors require cross-presentation of tumor-associated antigens (TAAs) by cDC1 especially migratory CD103$^+$ and lymphoid CD8α$^+$ DCs[46–49]. Although rare in TME, migratory CD103$^+$ and lymphoid CD8α$^+$ DCs can be recruited into TME and TDLNs by i.t. injection of Flt3L for nine consecutive days, as shown in previous studies[3,50]. Our study demonstrated a similar effect but reduced the dosing frequency. Moreover, both Flt3L and Lactis can stimulate NK cells, which will reversely produce the cDC1 chemoattractants CCL5 and XCL1, promoting cDC1 accumulation in mouse tumors[18,24]. T cells might be dysfunctional and exhausted with PD1 upregulation, along with the immune activation. Among them, in situ vaccine with Flt3L/ radiotherapy/ TLR3 agonists had been demonstrated to upregulate PD1 expression on T cells by almost two-fold in the tumor microenvironment[3]. As a potent immune adjuvant, FOLactis also significantly sensitized 'cold' tumors to immune checkpoint blockade therapy in our study. In addition to many attempts being made to block immune checkpoints, costimulatory molecules such as OX40, making a significant difference to the proliferation and survival of T cells, are also a promising strategy to further rejuvenate CD8$^+$ T cells to elicit durable antigen-specific T-cell responses[51,52]. Given that the synergistic therapeutic effect between locally injected TLR agonists and anti-OX40 antibodies has been explained in several studies[26], we hypothesized and proved the rational combination of Lactis/ OX40. We found that local delivery of FOLactis had the function of three elements (Lactis/ Flt3L/ OX40L) simultaneously, and it could effectively improve TME, converting 'cold' tumors to 'hot'.

Our results proved that ISV with FOLactis increased cross-presentation drastically due to the higher proportion of activated DCs especially CD103$^+$ and CD8$^+$ DCs in TME and TDLNs, which further induced CD8$^+$ T cell antitumour immunity and led to abscopal tumor regressions. The strategy was also shown to deflect M2 to M1 macrophages, affect other immune cells such as Tregs and NK cells in the microenvironment of treated tumors, and increase several important cytokines. Moreover, the therapeutic effect was specific for antigens expressed by the tumor at the injected site and could form a long-term memory in CT26-tumor-bearing mice, depending to a large degree on the activation of both innate and adaptive immune responses. In 4T1-tumor-bearing mice, FOLactis could greatly reverse the poor TME that was insensitive to immune effector mechanisms, with up-regulation of CD8$^+$ T cells and PD1$^+$ T cells. In B16F10-OVA-tumor-bearing mice, FOLactis also increased the frequency of OVA-tetramer$^+$ CD8$^+$ CTLs, initiating tumor-specific immune responses and eliciting systemic tumor regressions.

There are some advantages for potential clinical translation of the ISV. *Lactococcus lactis* has a favorable biosafety profile and there are

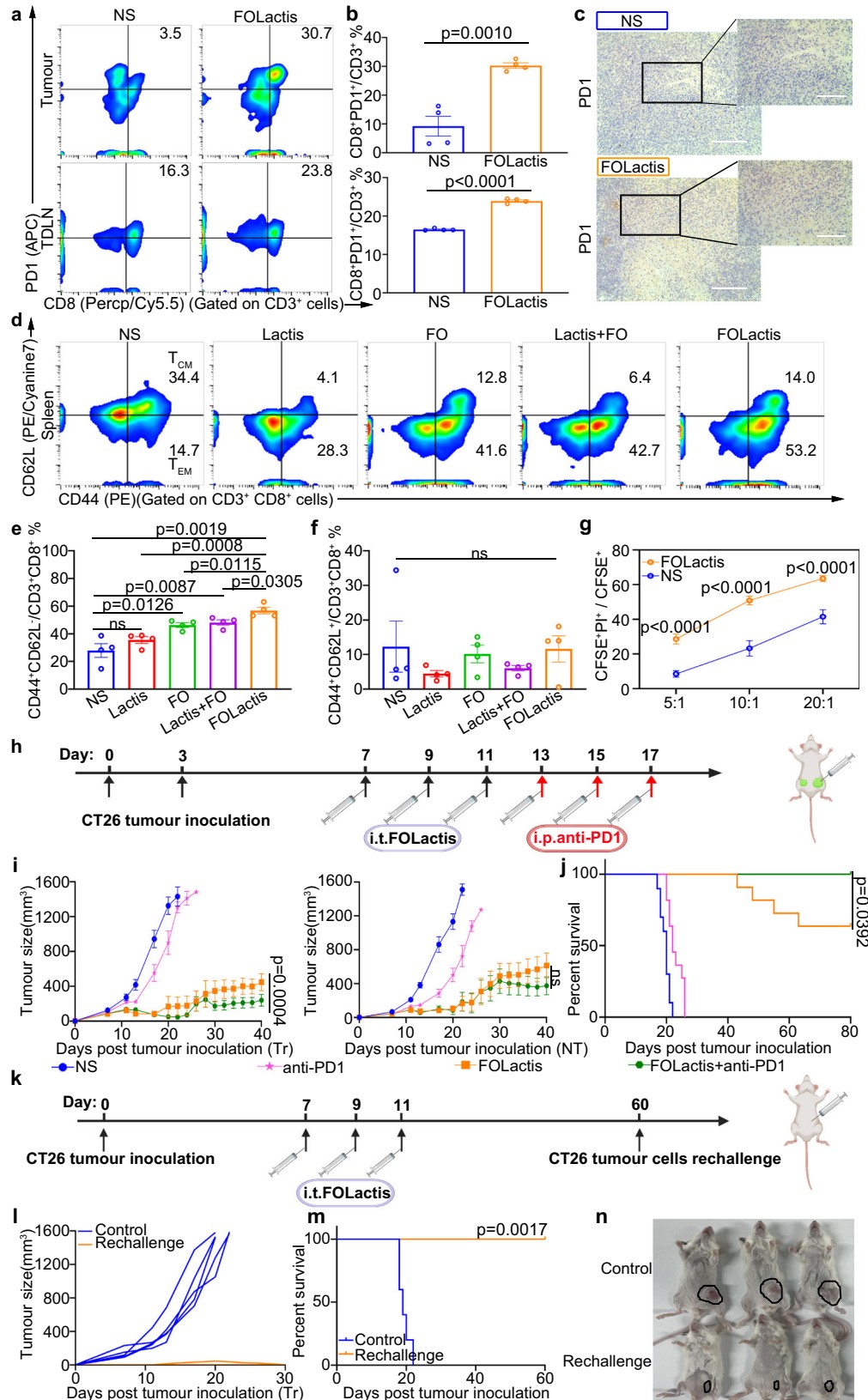

many clinical trials involving it. Meanwhile, the engineered probiotic can be readily produced from large-scale cultures at a relatively low cost, facilitating the production of antitumour vaccines. Granulocyte-macrophage colony-stimulating factor (GM-CSF) and Flt3L are both essential to the differentiation and survival of DCs. With the development of synthetic biology, the first FDA-approved local immunotherapy, the oncolytic viral therapy talimogene laherparepvec (TVEC) which delivered GM-CSF, has achieved inspiring success in clinical trials. Similarly, FOLactis could lead to continuous activation of immune cells and showed excellent tumor suppressive effect in mouse tumor models. Moreover, *Lactococcus lactis* can be modified to produce other significant immunostimulatory molecules such as

**Fig. 7 | ISV with FOLactis formed long-term immunological memory in a syngeneic tumor model and synergized with anti-PD1 antibody.** Representative flow cytometry images of PD1⁺CD8⁺ T cells seven days after treatments in CT26 treated tumors (**a**) and their flow cytometric analysis (**b**) ($n = 4$). **c** Immunohistochemistry analysis of PD1 expression in CT26 treated tumors seven days after treatments ($n = 3$). Scale bars, Left, 200 μm; Right, 100 μm. **d** Representative flow cytometry images of effector memory T cells ($T_{EM}$, CD3⁺CD8⁺CD44⁺CD62L⁻) and central memory T cells ($T_{CM}$, CD3⁺CD8⁺CD44⁺CD62L⁺) seven days after treatments in the spleen ($n = 4$) and their flow cytometric analysis (**e**, **f**). **g** Splenocytes of mice in the NS or FOLactis group were incubated with CFSE labeled CT26 cells at effector-to-target ratio (E: T) of 5:1, 10:1, and 20:1. PI was added six hours after incubation and the percentage of dead cells was analyzed by flow cytometry ($n = 3$). **h** Schematic diagram of administration route for FOLactis combined with anti-PD1 in tumor suppression experiment. Created with BioRender.com. Average tumor growth curves ($n = 6$) (**i**) and survival data ($n = 10$) (**j**) of tumor-bearing mice. The dose

of PD1 was 100 μg for each mouse in every administration. Tr: treated; NT: non-treated. **k** Schematic illustration of evaluating the immune memory effect. Mice cured of CT26 tumor were rechallenged subcutaneously (s.c.) 60 days later with $5 \times 10^5$ tumor cells. Created with BioRender.com. **l** Tumor growth curves of each mouse following the rechallenge ($n = 5$). **m** Survival data of two groups of mice ($n = 5$). **n** Pictures of representative mice were taken 15 days after rechallenge. The tumors of each mouse were circled. For the experiments in **b**, **e**, **f**, and **g**, the error bars represented mean ± s.e.m. statistical significance was calculated by two-tailed unpaired Student's t-tests. ns represented $p > 0.05$. For the experiments in **i**, data were the mean ± s.e.m. $p$-values were determined by two-way ANOVA with Tukey's multiple comparisons test. ns represented $p > 0.05$. Differences in survival were determined by using the Kaplan–Meier method, and the $p$ value was determined via the log-rank (Mantel–Cox) test. Source data are provided as a Source Data file.

tumor-specific antigens. Considering patient compliance, we also plan to try different delivery methods of FOLactis, such as oral or in situ hydrogel administration, which might reduce administration trauma and elicit long-term benefits. Given the unique characteristics of the bacterial membrane, such as negative charge or the lipid bilayer, we can modify them with nanometer materials, innovating a more intelligent 'Bacterial Robot' with multiple functions.

Some limitations still existed in our study. Several tumor-bearing mice might relapse four weeks after the treatment especially in 'cold' tumor models. Therefore, it is meaningful for us to deeply study the molecular perspective of this phenomenon and provide some more effective strategies such as adjusting the frequency and dose of administration and combining FOLactis with other effective immunotherapies. Moreover, it might be worthwhile to further study the changes of immune cells in nontreated tumors. If necessary, single-cell transcriptomic may be performed to analyze differences at the genetic level, and thoroughly uncover the development and metabolism of tumors.

In summary, we described a strong rationale for using FOLactis as an ISV. We believe that FOLactis represents a universal vaccine platform to provoke robust innate and tumor-specific adaptive immune responses for a broad range of solid tumors.

## Methods
### Reagents, bacterial strains, cell lines, and mice
*Lactococcus lactis* NZ9000 and pNZ8148 vector were obtained from MoBiTec (Germany). *Escherichia coli* BL21 (DE3) was purchased from Shanghai Weidi Biotechnology Co, Ltd. *Escherichia coli* strains were incubated in Luria-Bertani (LB) medium at 37 °C with shaking at 220 rpm. Lactis NZ9000 was propagated statically in M17 (Difco) medium containing 0.5% (w/v) glucose (GM17) at 30 °C. B16F10, 4T1, and CT26 cells were obtained from the Cell Bank of Shanghai Institute of Biochemistry and Cell Biology. CT26 / CT26-GFP-Luc colon cancer cells, MC38-GFP-Luc colon cancer cells, 4T1-GFP-Luc breast cancer cells and B16F10-OVA melanoma cells were cultured in Roswell Park Memorial Institute (RPMI) 1640 (Gibco) supplemented with 10% fetal calf serum, 100 U ml⁻¹ penicillin, and 100 μg ml⁻¹ streptomycins at 37 °C and 5% CO₂. Cells were tested for Mycoplasma, and only Mycoplasma-free cells were used. BALB/c and C57BL/6 male/female mice aged 5-6 weeks were purchased from Shanghai Sippr-BK laboratory animal Co. Ltd. (Shanghai, China) and kept in the specific pathogen-free (SPF) Laboratory Animal Center of Affiliated Nanjing Drum Tower Hospital of Nanjing University Medical School. All mice were maintained in specific pathogen-free animal facility with controlled temperature (68–79 °F), humidity (30–70%), and light/dark cycle (lights between 6 am and 6 pm). Feed and water were available ad libitum. All animal experimental protocols were approved by the Laboratory Animal Care and Use Committee of the Affiliated Nanjing Drum Tower Hospital of Nanjing University Medical School.

### Generation and characterization of FO and FOLactis
Flt3L-OX40L was designed in the following orientation: extracellular domain of Flt3L-PVGLIG-extracellular domain of OX40L. The DNA fragments were synthesized, cloned into bacterial expression vector pET28a or pNZ8148 by ICarTab Biomedical Co., Ltd. (Suzhou, China), and confirmed by enzyme digestion and DNA sequencing.

FO was expressed in *Escherichia coli* BL21 (DE3) after induction by IPTG. Bacterial cultures were harvested by centrifugation, resuspended, and disrupted by sonication. Inclusion bodies from the precipitation of ultrasonic lysates were solubilized in 8 M urea and dialyzed against at least 50-fold volume of renaturation buffer (6 M-4 M-2 M urea, pH 8.0) at 4 °C for 16–24 h with rapid mixing, followed by dialysis with only phosphate-buffered saline (PBS) for 24–48 h at 4 °C. The products were subjected to 12% sodium dodecyl sulfate-polyacrylamide gel electrophoresis (SDS-PAGE) analysis for confirmation. Refolded FO was purified using an ÄKTA system HisTrap HP column (GE healthcare, CT, USA) according to the manufacturer's instructions. The eluted solutions were dialyzed against PBS, sterile-filtered (0.22 μm), and confirmed by WB analysis using an anti-Flt3L antibody (Abcam ab231192). Protein concentrations were determined using a BCA assay kit (Thermo Fisher Scientific, Massachusetts, USA).

For FOLactis production, an overnight culture of Lactis NZ9000 was inoculated at 1:50 in fresh GM17 broth containing 1% glycine and 0.5 M sucrose and incubated at 30 °C to prepare competent cell. The culture was harvested by centrifugation (5000 × g, 4 °C, and 15 min) at the early exponential phase (OD600 = 0.3–0.5). The pellet was washed twice with an ice-cold washing solution I (10% glycerol, 0.5 M sucrose) and once with an ice-cold washing solution II (10% glycerol, 0.5 M sucrose, 0.05 M EDTA), resuspended in 1/100 culture volume of the ice-cold washing solution I, and then electroporated immediately or stored in aliquots at −80 °C. For electroporation, 40 μl of prepared competent cell was mixed with 1 μg of recombinant plasmid dissolved in double-distilled water, and transferred to an ice-cooled electroporation cuvette (0.2 mm). A single pulse was delivered by a Gene Pulser (Bio-Rad, USA) at 2.5 kV, 200 Ω. Immediately, the cell suspension was mixed with 1000 μl of recovery medium (GM17 broth containing 0.5 M sucrose, 20 mM MgCl₂ and 20 mM CaCl₂) and incubated at 30 °C for 2 h. After that, 200 μl of suspension culture was spread on a GM17 plate containing 10 μg ml⁻¹ chloramphenicol and incubated for 2 days. Then FOLactis was successfully prepared. For induction, FOLactis were cultured overnight in GM17 broth with 10 μg ml⁻¹ chloramphenicol and then inoculated 1:100 in fresh GM17 medium. When the OD600 of the culture reached 0.5–0.7, 10 ng ml⁻¹ nisin was added and cultured for an additional 3 h. Then, the bacterial cells were collected by centrifugation (5000 × g, 4 °C and 15 min) for protein expression analysis. After being treated with an ultrasound generator (Sonics VCX 130, USA; 130 W, 20 kHz) on ice for 6 min (on and off cycle of 5 s at 40% amplitude), the cell lysates were collected by centrifugation (12000 × g, 4 °C and 5 min). The supernatant of the cell lysates was added with equal volumes of

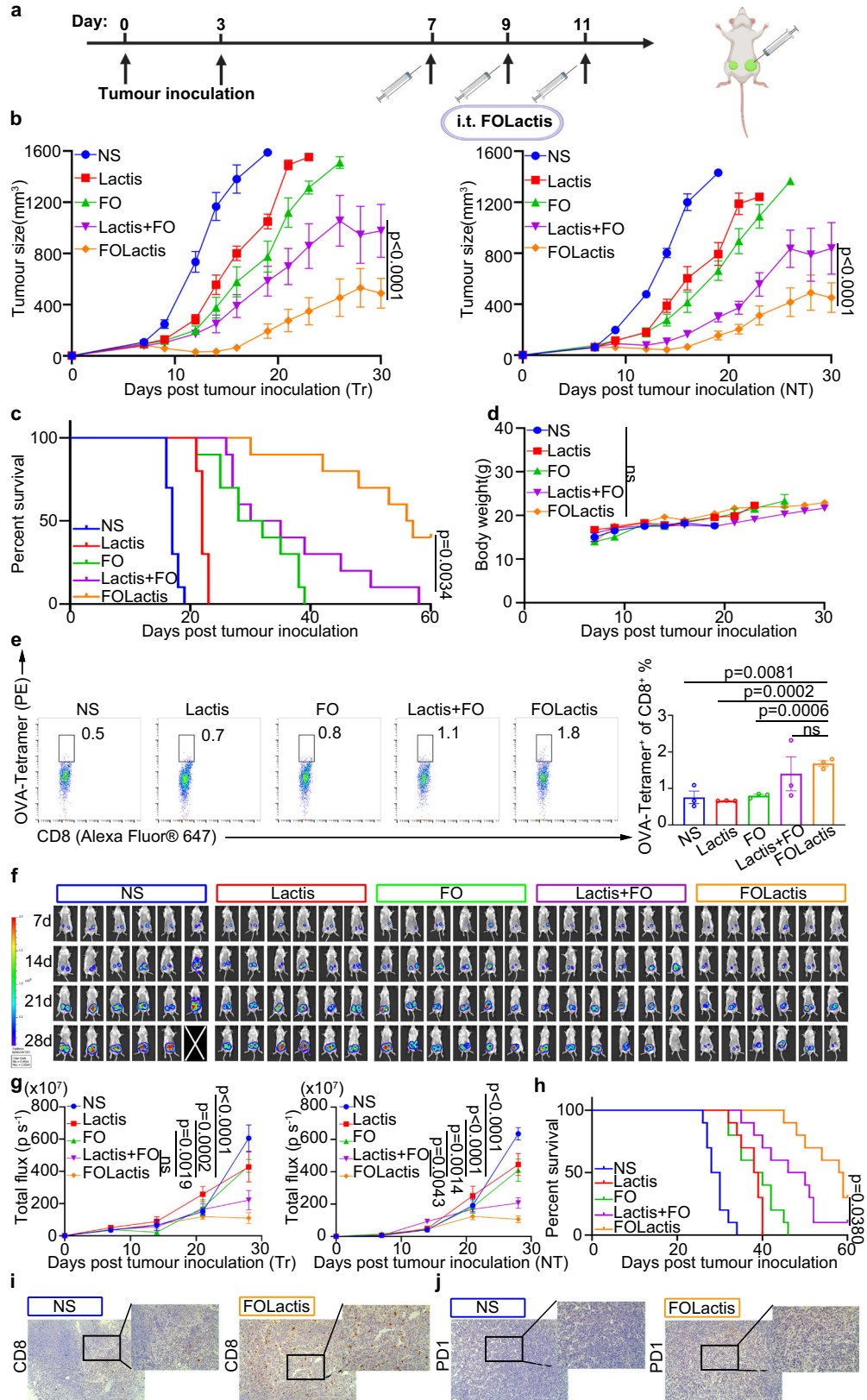

5× loading buffer and subjected to WB using an anti-Flt3L antibody (Abcam ab231192). According to the positive band in the PVDF membrane, the relevant band in polyacrylamide gels was excised, reduced and alkylated by 10 mM dithiothreitol and 55 mM iodoacetamide for protein identification using MS by Shanghai Applied Protein Technology (Shanghai, China). Target protein amounts in the bacterial lysates of FOLactis were assessed by ELISA (Mouse/Rat Flt3L Quantikine ELISA Kit, R&D Systems; Mouse OX40L ELISA Kit, AMEKO). ELISA results were recorded with SpectraMax® i3x multi-mode microplate reader (Molecular Devices) using SoftMax® Pro 6 software (version 6.4.2). FLactis, OLactis and FOLactis-sfGFP were constructed respectively using the similar methods as FOLactis. FLactis and OLactis expressing

**Fig. 8 | ISV with FOLactis initiated antigen-specific therapeutic effect and inhibited multiple types of immunologically "cold" cancer. a** Schematic diagram of the therapeutic treatment in tumor-bearing mice. Mice were inoculated (s.c.) with tumor cells ($5 \times 10^5$ per mouse), and received treatments on days 7, 9 and 11. Created with BioRender.com. Average tumor-growth curves (**b**), survival data (**c**) and body weight (**d**) of C57BL/6 mice bearing B16F10-OVA tumor with different treatments as indicated ($n = 8$). The dose of different treatments was the same as Fig. 4b. Tr: treated; NT: nontreated. **e** Generation of antigen-specific CD8[+] T cell responses in tumors of B16F10-OVA-bearing mice ($n = 3$). Representative flow cytometry images of OVA[+]CD8[+] T cells gated on CD3[+] T cells two days after treatments and their flow cytometric analysis. **f–h** BALB/c mice were implanted with 4T1-GFP-Luc cells ($2 \times 10^5$) on the left and right lower sides of the abdomen on day 0 or day 3, and received treatments on days 7, 9, and 11 ($n = 6$). Anterior bioluminescence images of tumor burden on days 7, 14, 21, and 28 after tumor inoculation (**f**). Shown are the tumor signal (**g**) and overall survival (**h**). **i, j** Immunohistochemistry analysis of CD8 or PD1 expression in 4T1-GFP-Luc treated tumors seven days after treatments ($n = 3$). Scale bars, Left, 200 μm; Right, 100 μm. For the experiments in **b** and **g**, data were the mean ± s.e.m. $p$-values were determined by two-way ANOVA with Tukey's multiple comparisons test. ns represented $p > 0.05$. For the experiments in **d** and **e**, data were the mean ± s.e.m. $p$-values were determined by two-tailed unpaired Student's t-tests. ns represented $p > 0.05$. Differences in survival were estimated by the Kaplan–Meier analysis, and the $p$ value was determined via the log-rank (Mantel–Cox) test. Source data are provided as a Source Data file.

protein was confirmed by WB analysis using an anti-His antibody (Cell Signaling Technology #12698S). All antibodies were diluted 1:1000.

### In vitro BMDCs uptake of FOLactis

BMDCs were obtained from bone mesenchymal stem cells harvested from C57BL/6 mice freshly. Then the cells were cultured with RPMI 1640 medium containing 10% FBS and 1% penicillin-streptomycin as described in the previous literature[53]. We added 20 ng ml⁻¹ rmGM-CSF (Xiamen Amoytop Biotech Co., Ltd., China) and 10 ng ml⁻¹ rmIL-4 (Pepro Tech, USA) to the medium to induce the cells to differentiate into DCs. The medium was replaced every three days and these DCs were centrifuged, and collected for use on day 8. FOLactis stained by DiO (Bridgen, Beijing, China) were co-incubated with DCs stained by DiI (Bridgen, Beijing, China) for two hours and imaged under a confocal laser scanning microscope (Leica, Germany) to assess the process of internalization into DCs.

**Western blot analysis of BMDCs.** BMDCs protein was extracted using the Nuclear and Cytoplasmic Protein Extraction Kit (Cat. P0027) and the RIPA Lysis Buffer (Cat. P0013C) from Beyotime Institute of Biotechnology (China). Proteins were confirmed by WB analysis using an anti-TLR1 antibody (Abcam, ab37068), anti-TLR2 antibody (Abcam, ab209216), anti-TLR6 antibody (Cell Signaling Technology #12717), β-actin antibody (Cell Signaling Technology #4967), anti-Phospho-NF-κB p65 (Cell Signaling Technology #3033S) and anti-Histone H3 (Cell Signaling Technology #4499S). All antibodies were diluted 1:1000.

### In vitro release of the fusion protein from the engineered Lactis

$10^9$ CFU FOLactis-sfGFP were dissolved in 500 μl PBS with/without BMDCs, splenocytes or CT26 tumor cells at 37 °C. At indicated time points, we obtained the supernatant by centrifugation ($12000 \times g$, 4 °C and 5 min) and detected the fluorescence intensity of sfGFP using a Varioskan Lux microplate reader (Thermo Fisher Scientific). We extracted and purified FO-sfGFP from the lysates of $10^9$ CFU FOLactis-sfGFP as the total protein. The release rate of protein was calculated according to the following formula:

$$\text{Release of protein}\,(\%) = \text{Fluorescence intensity of FO} - \text{sfGFP/Fluorescence intensity of total FO} - \text{sfGFP}$$

### In vitro BMDCs stimulation

Effects of different concentrations of Live Lactis or FOLactis, FO and the crude lysates from Lactis or FOLactis on BMDCs were studied in vitro by co-culture for 24 h. Then DCs were collected by centrifugation at $500 \times g$ for 5 min and incubated with fluorescein isothiocyanate (FITC)-CD11c, allophycocyanin (APC)-CD80, and phycoerythrin (PE)-CD86 for 30 min before evaluation.

### Preparation of FOLactis lysates

FOLactis were cultured in liquid GM17 medium at 30 °C without shaking. When the OD600 of the culture reached 0.5–0.7, 10 ng ml⁻¹ nisin was added and cultured for an additional 3 h. Then, the bacterial cells were collected by centrifugation ($5000 \times g$, 4 °C and 15 min). The cell pellet was resuspended in PBS and treated with ultrasound generator (Sonics VCX 130, USA; 130 W, 20 kHz) on ice for 30 min (on and off cycle of 5 s at 40% amplitude). The resulting lysate ($10^{10}$ CFU ml⁻¹) was filtered through a 0.2 μm filter, frozen and lyophilized in a freeze dryer to obtain the FOLactis lysates.

### Preparation of TM

The murine CT26 cells were collected in logarithmic growth phase and the membranes were extracted as shown in the previous literature[54]. Briefly, the cells ($1 \times 10^8$ cells ml⁻¹) were lysed by freeze-thaw cycling (10 min in liquid nitrogen and 10 min in 37 °C water bath) for at least six times. Then the precipitation was removed by centrifuging cell lysates at $700 \times g$ for 10 min. After being treated with ultrasound generator (Sonics VCX 130, USA; 130 W, 20 kHz) on ice for 2 min (on and off cycle of 5 s at 50% amplitude), the CT26 TM was collected by centrifugation ($15000 \times g$, 4 °C and 30 min) and stored in −80 °C for further study.

### In vitro T-cell activation

Lymphocytes were isolated from splenocytes of BALB/c mice to co-culture with 50 μg TM, 50 μg TM + 50 μg FO, 50 μg TM + $10^7$ Lactis lysates or 50 μg TM + $10^7$ FOLactis lysates regarding the in vitro stimulation method of long peptides[55]. Briefly, we used lysis buffer (Invitrogen eBioscience) to obtain cells in the spleen except for the red blood cells and cultured them in AIMV medium containing 40 ng ml⁻¹ rmGM-CSF, 20 ng ml⁻¹ rmIL-4, and 0.5% FBS. The cells were divided into 96-well plates ($5 \times 10^6$ cells per well in 100 μl), and the different drugs above were added to each well on day 1, with 0.3 μg resiquimod (R848, BIOFOUNT, China) and 5 ng LPS 4 h or 4.5 h later. All the cells were collected on day 2 and resuspended in AIMV medium containing 10% FBS, 1% P/S, 50 ng ml⁻¹ IL-7, and 24 IU ml⁻¹ IL-2 (Pepro Tech, USA). Then we replaced the medium in half every 2–3 days until day 10. Meanwhile, follow the steps in 0–2 days to prepare new cells on day 8 and add them to the previous cells to co-culture for 24 h. Finally, the cells were collected by centrifugation and assessed by flow cytometry analysis of anti-CD4-PE, anti-CD8-APC, anti-CD69-PE/Cyanine7, and anti-CD25-FITC.

### In vivo real-time near-infrared fluorescence imaging

Near-infrared imaging was used to localize FOLactis quantitatively. FOLactis labeled with DiR (Bridgen, Beijing, China) were intratumorally injected. Then we used CRi Maestro In Vivo Imaging System (Cambridge Research & Instrumentation, Massachusetts, USA) to anesthetize and scan the mice at indicated time points post FOLactis administration. Tumors, TDLNs, and central organs, including heart, liver, spleen, lung, and kidney were excised and imaged for resected tissue imaging after the mice were sacrificed under deep anesthesia.

**Bacterial colonization in vivo.** After intratumoural injection of FOLactis in the treated tumor, we collected the major organs, blood and tumor tissue at desired time points. Then we weighed and

homogenized these samples at 4 °C in sterile PBS (pH = 7.2). Homogenates were serially diluted and plated on GM17 plates at 30 °C for 48 h. Bacterial colonies were counted and computed as CFU/g of tissue.

## Immunofluorescence confocal imaging

To further delineate the trajectories of FOLactis in vivo, we harvested tumors, TDLNs, and main organs, including the heart, liver, spleen, lung, and kidney, after i.t. injection of DiO-labeled FOLactis at the required time. After washing with PBS, the frozen sections were mounted with DAPI (Beyotime, Shanghai, China) and then imaged on a confocal laser scanning microscopy (Leica, Germany). To evaluate the internalization of FOLactis into DCs, DiO-labeled FOLactis ($1 \times 10^9$ CFU per mouse) were intratumourally injected and tumors were excised 24 h later. We prepared tumor slices, blocked these tumor sections with 2% BSA for 30 min, and stained them with anti-CD11c-PE antibody and anti-CD103-APC antibody overnight. Finally, the tumor slices were imaged using a confocal laser scanning microscope (Leica, Germany). The fluorescence intensity of CD11c$^+$ CD103$^+$ DCs was statistically analyzed using Image-Pro Plus 6.0.

## Immunohistochemistry analysis

To evaluate the changes in immunogenic microenvironment in treated tumors, we collected tumors seven days after the last treatment and incubated paraformaldehyde-fixed tumor tissue sections with primary antibodies (CD8, PD1) for 2 h at 37 °C and then washed them with PBS three times. CD8 (EPR21769, Abcam) and PD1 (EPR20665, Abcam) were purchased from Abcam company. Image-Pro Plus 6.0 was adopted to analyze each, and mean density (IOD/area) was used to analyze protein expression.

## Tumor inoculation and i.t. therapy for subcutaneous models

$5 \times 10^5$ CT26 tumor cells were injected (s.c.) in 100 µl NS on the left lower sides of the abdomen for single-tumor efficacy experiments. For two-tumor models, the same number of cells ($5 \times 10^5$ CT26, $2 \times 10^5$ 4T1-GFP-Luc, $1 \times 10^5$ B16F10-OVA) was inoculated on the contralateral abdomen three days after the primary tumor. Tumors were treated intratumourally with NS, $10^9$ CFU Lactis, 50 µg FO, $10^9$ CFU Lactis + 50 µg FO or $10^9$ CFU FOLactis on days 7, 9, and 11, which were dissolved in NS to a final volume of 100 µl per dose. For the dose fumbling experiment, different concentrations of FO or FOLactis were injected intratumourally as described above. Tumor volume was measured every 2-3 days, calculated by the formula length × width$^2$ × 0.5. Maximal tumor burden permitted was 1500 mm$^3$. In some cases, this limit has been exceeded the last day of measurement and the mice were immediately euthanized. Central organs, including the heart, livers, spleen, lung, and kidneys were harvested, fixed in 4% paraformaldehyde, sectioned, and stained with H&E for safety analysis under optical microscopy (DM5000, Leica, Germany). Blood samples (-1.0 ml for each mouse) were also collected for blood biochemistry analysis to measure serum cytokines using LEGENDplex™ MU Th1/Th2 Panel (8-plex) w/ VbP V03 (Biolegend).

For anti-PD1 immunotherapy, CT26 tumor-bearing mice were randomly divided into four groups (n = 6): (i) NS, (ii) FOLactis, (iii) anti-PD1, (iv) FOLactis and anti-PD1. The anti-PD1 (100 µg per mouse for each injection) was administrated (i.p.) on days 13, 15, and 17 after the last injection of FOLactis.

In the 4T1-GFP-Luc mouse tumor models, mice were randomly divided into five groups for different treatments. Tumor burden was monitored on days 7, 14, 21, and 28 after tumor inoculation using the IVIS Lumina III system (PerkinElmer, Massachusetts, USA).

## Flow cytometry

Antibodies to CD11c (N418, FITC, 117306), CD11c (N418, PE, 117308), CD80 (16-10A1, APC, 104714), CD86 (GL-1, PE, 105008), CD8a (53-6.7,

PerCP/Cyanine5.5, 100734), CD103 (2E7, PE, 121406), CD3 (500A2, FITC, 152304), PD1 (29F.1A12, APC, 135210), CD4 (GK1.5, PE, 100408), CD8a (53-6.7, APC, 100712), CD25 (PC61, FITC, 102006), CD25 (PC61, APC, 102012), CD69 (H1.2F3, PE/Cyanine7, 104512), CD69 (H1.2F3, FITC, 104506), CD4 (GK1.5, PE/Cyanine7, 100422), CD25 (PC61, APC, 102012), FoxP3 (MF-14, PE, 126404), CD44 (IM7, PE, 103008), CD62L (MEL-14, PE/Cyanine7, 104418), CD11b (M1/70, APC, 101212), F4/80 (BM8, PE/Cyanine5, 123112), CD206 (C068C2, PE, 141706), CD134 (OX-86, APC, 119414), CD49b (DX5, PE, 108908), CD135 (A2F10, APC, 135310), IFN-γ (XMG1.2, APC, 505810) were purchased from Biolegend. Anti-CD8 (Mouse) mAb-Alexa Fluor® 647 (KT15) was obtained from MBL. All antibodies were diluted 1:100. H-2K$^b$ OVA Tetramer-SIINFEKL (TS-5001-1C) was purchased from MBL and performed according to the instructions. Intracellular staining for FoxP3 was performed using the True-Nuclear™ Transcription Factor Buffer Set (Biolegend).

Tumor tissues, TDLNs, and spleens were taken out from mice. Single-cell suspension from the spleen and TDLNs was prepared using the mechanical trituration method while tumor tissues minced into small pieces were digested with collagenase type IV (1 mg ml$^{-1}$, Sigma) for 2 h at 37 °C with gentle agitation. All samples were then resuspended in ice-cold NS, stained with specific antibodies for 20 min in 4 °C in darks, and washed before analysis. LEGENDplex™ MU Th1/Th2 Panel (8-plex) w/ VbP V03 (Biolegend) was used to detect and analyze the level of cytokines in the tumor cell supernatant or serum. Cells were analyzed using BD Accuri C6 (BD Bioscience, USA) and analyzed by FlowJo software.

## Mouse tumor RNA sequencing and gene expression analysis

Mice were killed 2 days after the last treatment. We quickly excised tumors and frozen them with liquid nitrogen. The mRNA samples of the NS group and FOLactis group were used for RNA-seq (GENEWIZ, Suzhou, China). Quantification of gene expression and differential expression analysis was performed using DESeq2 Bioconductor package. Functionally related GO terms for biological processes were analyzed by GOSeq (v1.34.1), while Kyoto Encyclopedia of Genes and Genomes (KEGG) enrichment analysis used the database (http://en.wikipedia.org/wiki/KEGG). GO network analysis of significantly upregulated genes in tumors were analyzed by Cytoscape software.

## Depletion studies

For depletion of immune cell subsets, mice were injected (i.p.) with anti-mouse CD8 (clone 2.43, BioXCell, 400 µg per injection twice weekly), anti-mouse CD4 (clone GK1.5, BioXCell, 200 µg per injection weekly), anti-mouse CSF1R (CD115, BioXCell, 300 µg per injection every other day) and anti-ASGM1 (anti-mouse Asialo-GM1, 50 µl per injection twice weekly) as previously described[26,56,57].

## Statistics and reproducibility

Measurements in this study were obtained from distinct samples. Statistics and graphs were performed using GraphPad Prism 8 (San Diego, CA, USA). Graphs include means and error bars, with all results presented as means ± s.e.m. as indicated. For survival studies, log-rank (Mantel−Cox) tests were used. For tumor burden comparisons, two-way analyses of variance (ANOVAs) followed by Tukey's multiple comparison tests were used. For FACS studies and other experiments, student's t-tests were used. All experiments were repeated at least three times. Flow cytometry data were collected with BD Accuri C6 (BD Bioscience, USA) and analyzed using FlowJo. Figures were designed in Adobe Photoshop.

## Reporting summary

Further information on research design is available in the Nature Portfolio Reporting Summary linked to this article.

## Data availability

Source data are provided as a Source Data file. The RNA-seq data generated in this study have been deposited in the GSA database under accession code CRA008825. The remaining data are available within the Article, Supplementary Information or Source Data file. Source data are provided with this paper.

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

## Acknowledgements

J.Z., Y.K., and Q.L. contributed equally to this work. This work was financially supported by the National Natural Science Foundation of China (81930080 (B.L.), 81972309 (Q.L.) and 82072926 (F.M.)), and the fundings for Clinical Trials from the Affiliated Drum Tower Hospital, Medical School of Nanjing University.

## Author contributions

J.Z. and B.L. conceived and designed the experiments. J.Z., Y.K., and Q.L. performed the experiments. F.L., R.X., H.Z., A.C., J.X., F.M., J.W., R.L., and L.Y. assisted in the experiments and data analysis. J.Z., Y.K., Q.L., and J.Y. prepared the manuscript. B.L. supervised the project.

## Competing interests

The authors declare no competing interests.
