## [Peer Review File · Nature Communications]

Reviewers' Comments:

Reviewer #1:

Remarks to the Author:

The manuscript entitled "Bifunctional engineered *Lactococcus lactis*-based in situ vaccination elicits systemic tumour regressions" describes *Lactococcus lactis* expressing a fusion protein of Fms-like tyrosine kinase 3 ligand and co-stimulator OX40 ligand (FOLactis). Intratumoral delivery of FOLactis demonstrated an enhanced antitumor immunity and conversion of cold tumors to hot tumors.

The study starts from interesting idea and design, and the manuscript is written well. However, we should address several major points before publication.

1. In general, in vivo studies are done in small number of animals. Mostly n=3 or 4. Authors should increase the number of animals.
2. The way of producing FO from *Lactococcus lactis* is not clear. In graphics of Fig. 1a FOLactis seems to secrete FO ligands in situ. However, in Fig. 1b, Flt3L-OX40 (FO) fusion protein was identified by FOLactis lysates, while some in vitro immunological assays was done using FOLactis lysate (Fig 2b, d-h). In order for the FO fusion protein carried by live *Lactococcus* to produce an anticancer effect, the protein must be well secreted out of the bacteria. But authors did not verify the solubility and secretion of fusion protein from *Lactococcus lactis*. And all in vitro study should also be done using live FOLactis.
3. The action mechanism of ligands that acts with the receptor of immune cells and causes immune response has not been verified. In the manuscript, authors showed internalization of FOLactis by DC (Fig. 2a), but did not demonstrate downstream signal pathway.
4. In Fig. 3, authors demonstrated biodistribution of FOLactis using DiO labeled bacteria until 360 h after i.t. injection. FOLactis will overgrow in tumor and other tissue when they colonize. DiO signal will weaken as bacteria divide. How can you quantitatively estimate the amount of bacteria in many tissues using this method. I suggest the enumeration of FOLactis counts in different organs at different time points after injection.
5. Authors determined the concentration of FO 50 ug in vivo based on the antitumor efficacy study in Supplementary Fig. 6. The rationale of comparing FOLactis 109 CFU with FO 50 ug + Lactis 109 CFU is not understandable (Fig. 4b). How much amount of FO can be produced from 109 CFU FOLactis? If it is over 50 ug, why don't you compare with 100 ug FO + 109 CFU Lactis? If therapeutic outcome of 100 ug FO + 109 CFU Lactis is better than that of 109 CFU FOLactis, what is the merit of 109 CFU FOLactis?
6. Authors suggested ISV effect of FOLactis based on the data in Fig. 4c. I am not sure that this can be suggested as ISV effect because authors didn't check the presence of FOLactis in the untreated tumor (opposite side). Tumor reduction in untreated tumor can be interpreted as moving from the location of treated tumor to the opposite site. Authors should count the bacterial number and evaluate the immunological phenotypes in the untreated tumors.
7. Authors did not assess the antitumor effect of Lactis expressing Flt3L only or OX40L only with that of FOLactis.
8. RNA-seq analysis of tumors after FOLactis treatment, authors observed activation in both innate and adaptive immunity through T cell activation and NK cell activation. I am so curious of the effect of Lactis itself whether Lactis can stimulate innate immunity or not.
9. Supplementary Fig 10 and 11 data showed biosafety of FOLactis on 7 dpi. Toxicity such as cytokine storm can occur earlier than 7 dpi. Serum study on 3 dpi would be needed.
10. The first paragraph of Discussion should move to the initial part of Introduction.

Reviewer #2:

Remarks to the Author:

This the first submmision of this manuscript, Zhu et. al., reporting that they developed a probiotic bacterium *Lactococcus lactis* expressing a fusion protein consisting of Fms-like tyrosine kinase 3 ligand and and co-stimulator OX40 ligand. This recombinant *L. lactus* was further evaluted as in situ vaccine. The vaccination regimen show significant efficacy in three different tumorm models, key components of the anti-tumor immune response were activated such as NK cells and CD8+. When combined with immune check point inhibitor anti-PD1 the anti-tumor efficacy was greater

than each agent separately. This is a novel study with a great potential to influence the field of probiotic immunotherapy. However, there are some weaknesses noted as follow:

1. The authors claim that the fusion protein FO is sustainably released. However, the construction did not include a secretion signal, how is the fusion protein released?
2. The author also claims that *L. lactis* expressing FO fusion protein could kill cancer cells by direct cell lysis. However, there is no data or experiments demonstrating such effect.
3. The authors hypothesize that *Lactococcus lactis* could increase the expression of OX40 on CD4+ T cells in TME by PAMPs. However, the reference 20 is not about pathogen-associated molecular patterns. It is not clear whether this is the case because *L. lactis* lacks of toxicity and immunogenicity. <https://www.frontiersin.org/articles/10.3389/fimmu.2017.01961/full>
4. The authors mentioned that they used orthotopic mouse models but in reality they used subcutaneous tumor models.
5. In the line 248 the authors used the word "obviously" this word is not acceptable in a scientific manuscript.

Reviewer #3:

Remarks to the Author:

In this work, the authors describe a new immunotherapy strategy based on the use of a bacterial vector harboring a chimeric protein which has two immunomodulatory domains: Flt3L and OX40L. The results presented are consistent, and very interesting, demonstrating the potential of using the strategy for the development of innovative clinical approaches. I have a few questions to discuss and also minor observations, that may contribute to the paper.

Questions to discuss:

1. Flt3L has been described to suppress GVHD associated to DCs signaling, inhibiting T cells (<https://doi.org/10.3389/fimmu.2021.699128>). On the other hand flt3L enhances DC expansion and activity that may contribute to anti-cancer therapies. Please discuss about switching this balance between immunotolerance and enhancing antitumor activity. How can you drive flt3L signaling to antitumor response?
2. Please comment about figure 4h-j, whether depletion of CD4 and CD8 cells includes both DCs and T cells. I'm curious if authors also tested anti-CD3 depletion.
3. Authors propose to use bacterial vectors expressing FO as a drug for cancer therapy. Since bacterial administration into sterile body compartments (as tumor site) may elicit immune response, it was shown several experiments to investigate adverse effects. These experiments were performed challenging animals with bacterial vector. I think it should be interesting to rechallenge animals and evaluating adverse effects. Another related question to discuss: what happens if these bacterial vectors enter blood stream?
4. When authors purified the FO, it was described a complex protocol for protein purification from inclusion bodies, that are expected as Flt3L and OX40L are eukaryotic membrane proteins. If these proteins should be purified like that, why does bacterial vector expressing FO secrete these recombinant protein? Please clarify.
5. Technical question 1: line 100: why is it important to have "extracellular domain of the Flt3L at the N-terminus fused to the extracellular domain of the OX40L at the C-terminus by a matrix metalloproteinase (MMP)-sensitive peptide Pro-Val-Gly-Leu-Iso-Gly"? Is it important to separate flt3L from OX40L for biological effect? I think you used same strategy for "FOlactis" (supp fig 1, supp fig 2)? Have you compared the *Lactis* + flt3L + OX40L with *Lactis* expressing a fusion protein flt3L-OX40L that hasn't MMP site? Is the fusion protein harboring MMP active only in the tumor site?
6. Technical question 2: the FO protein expressed in *E. coli* should be purified from inclusion bodies, on the other hand, authors mention that *Lactococcus lactis* may be engineered to produce a FO protein that is secreted, enhancing immunity. What is different in FO expression in *Lactis* that

FO protein doesn't go to inclusion bodies and is secreted?

Minor observations:

Abstract: I suggest reviewing English language. Egg Lines 28-29 a lot of "and"

Line 75 why does Injection of FOlactis could kill cancer cells by direct cell lysis? I couldn't find this explanation in the indicated references (14-16)

Line 88 - what's "multiple orthotopic" models were used?

Figure 1c: Elisa assay isn't clear. Please explain y axis and x axis. What does it means murine flt3L/OX40L (pg/mL-1)? Are you normalizing Flt3L per OX40L? - in the X axis, what does it means FOlactis:Flt3L?

Line 108: "According to the manufacturer's instructions" which manufacture? For BL21? For FO ?

Fig 2b caption, please describe concentration for FO, Lactis and FOlactis that correspond to the graph bars (FO: 1-100ug/ml, lactis and FOlactis: 10(5) to 10(8) cfu/mL).

Fig 7i: I think it should be written anti-PD1 instead PD1 in graph legend?

Caption of figure 7: there are squares instead characters line 441.

Suppl. Fig 1: I suggest clarifying to highlight sequences that are shown in "B", to match with the scheme "a": Flt3L sequence in orange, PVLIG sequence in blue and OX40 in red. What is pnis?

What are sequences labelled in RED in "b", that you describe as best matched peptides?

Supplementary fig 10 , caption , line 3: there are squares instead characters

Supplementary fig 11, I thing it should be interferon gamma (IFNg) instead "IFN-r" y axis.

Point-by-point responses to reviewers

Note: Following are our responses (in blue color) to reviewers' comments and sentences described in the revised manuscript (highlighted in yellow).

Reviewer #1 (Remarks to the Author): with expertise in bacteria-based cancer therapy, immunotherapy

The manuscript entitled “Bifunctional engineered *Lactococcus lactis*-based in situ vaccination elicits systemic tumour regressions” describes *Lactococcus lactis* expressing a fusion protein of Fms-like tyrosine kinase 3 ligand and co-stimulator OX40 ligand (FOLactis). Intratumoural delivery of FOLactis demonstrated an enhanced antitumour immunity and conversion of cold tumours to hot tumours. The study starts from interesting idea and design, and the manuscript is written well. However, we should address several major points before publication.

Comment 1:

1. In general, in vivo studies are done in small number of animals. Mostly n=3 or 4. Authors should increase the number of animals.

Answer 1:

Thank you very much for your comments. We have repeated the tumour suppression experiment in CT26 tumour-bearing mice and increased the sample size to 6 to make a more reliable conclusion. The conclusion remains the same as before. The results are shown in Figure 5b-i and Supplementary Fig.12. The legend for Figure 5b-i and Supplementary Fig.12. has also been revised. We have also revised the information of results and methods.

Comment 2:

2. The way of producing FO from *Lactococcus lactis* is not clear. In graphics of Fig. 1a, FOLactis seems to secrete FO ligands in situ. However, in Fig. 1b, Flt3L-OX40 (FO) fusion protein was identified by FOLactis lysates, while some in vitro

immunological assays was done using FOLactis lysate (Fig 2b, d-h). In order for the FO fusion protein carried by live Lactococcus to produce an anticancer effect, the protein must be well secreted out of the bacteria. But authors did not verify the solubility and secretion of fusion protein from Lactococcus lactis. And all in vitro study should also be done using live FOLactis.

Answer 2:

We are very grateful for your kind suggestions, and fully agree that it is of great value to evaluate whether the fusion protein can be well secreted out of the engineered lactis and retain its biological structure and activity. There are some other studies¹⁻³ using bacteria that express proteins intracellularly as potential therapeutics, which proves the feasibility of this strategy. When bacteria invade the body, they are attacked by immune system via three main ways: (i) via complement-mediated lysis^{4, 5}; (ii) via phagocytosis⁶⁻⁸; (iii) via cell-mediated immunity⁹.

According to these studies, we assume that the fusion protein will release as bacteria die. We constructed an engineered Lactis (FOLactis-sfGFP) expressing the fusion protein Flt3L-OX40L-sfGFP (added in Supplementary Fig.4). After the induction of nisin, FOLactis-sfGFP turned green, indicating that the expressed heterologous protein retained its biological structure and biological activity, which is an important characteristic for successful immunoregulation. Besides, we detected the release of the fusion proteins. FOLactis-sfGFP released the fusion protein in PBS slowly and continuously, while it took lesser time to release the fusion protein in PBS mixed with BMDCs or splenocytes (added in Supplementary Fig. 4c, e-f). The corresponding results have been revised in Supplementary Fig.4.

Based on your suggestions, we evaluated the effect of live FOLactis on DCs and T cells activation. We found that the activation effects of FOLactis on BMDCs were dose-dependent. There was a significant increase in the percentage of mature DCs (CD80⁺ CD86⁺) in the FOLactis group within the concentration range of 10⁷-10⁸ CFU

ml⁻¹, even much better than the LPS group (24.5%). When the concentration of Lactis reached 10⁷ CFU ml⁻¹, the average maturation proportion of BMDCs (29.7%) reached a peak. Splenocytes of the BALB/c mice were used as a source of DCs and T cells to induce activated or memory T cells. After treated with live Lactis or FOLactis for 24 hours in in vitro, we observed a higher expression of CD69 (early activation marker) and CD25 (late activation marker) in the FOLactis group than that in the Lactis group, leading to the robust generation of IFN- γ . In summary, our findings suggested that both FOLactis and their debris could act as immune stimulatory agents (added as Fig. 2b, e-f). We hope our revisions can address your concern.

Supplementary Fig.4 | Sustained release of the fusion protein from the engineered Lactis. **a**, Plasmid map of a Lactococcus lactis-Escherichia coli shuttle vector using pNZ8148 expressing the Fit3L-OX40L-sfGFP fusion protein. **b**, Cellular uptake of FOLactis-sfGFP after a two-hour incubation with BMDCs, as assessed by flow cytometry (n=3, biologically independent samples). **c**, 10^9 CFU FOLactis or FOLactis-sfGFP were resuspended in PBS at 37°C . In vitro images to demonstrate the activity and release of the Fit3L-OX40L-sfGFP

fusion protein. **d**, Colocalization analysis of FOLactis-sfGFP (green) with lysosomal compartments (red) in BMDCs by confocal microscopy (two-hour incubation). White scale bars, 10 μ m. **e-f**, Curves of the Flt3L-OX40L-sfGFP fusion protein release from the engineered Lactis with or without other cells in PBS at 37 ° C (n=5).

Fig.2 | FOLactis efficiently enhanced the activation of DCs and T cells, and induced tumour cell death in vitro. **a**, Colocalization analysis of FOLactis (DiO; green) in bone marrow-derived DCs (BMDCs) (DiI; red) by confocal microscopy (two-hour incubation). White scale bars: Up, 50 μm ; Down, 10 μm . **b**, The percentage of mature DCs (mDCs, CD11c⁺CD80⁺CD86⁺) after co-incubation with Lactis or FOLactis in vitro for 24 hours (n=6). **c**, Western blot analysis of several membrane TLR proteins and NF- κ B pathway in BMDCs. The gels were loaded with equal amounts of the proteins (10 μg). **d**, Pro inflammatory cytokine concentrations in BMDC supernatants (n=6). **e**, Assessment of IFN- γ in coculture supernatants after T cells from the spleen of BALB/c mouse stimulated by Lactis or FOLactis for 24 hours in vitro (n=6). **f**, The quantification of CD69 and CD25 expression on CD8⁺ and CD4⁺ T-cell subsets in the T cells (n=6). **g**, Luminescence was detected by the addition of D-luciferin to CT26-luc cells after incubated with T cells and/or FOLactis for 24 hours (n=6). Cells were examined under the microscope EVOS FL Auto Cell Imaging System (Invitrogen) and photographed. White scale bars, 200 μm . **h-i**, The Flt3L-OX40L-sfGFP fusion protein (FO-sfGFP) was extracted from lysates of FOLactis-sfGFP. Confocal laser scanning microscopy images of mouse BMDCs treated with FO-sfGFP and APC-anti mouse Flt3 for two hours. Confocal laser scanning microscopy images of mouse splenocytes treated with FO-sfGFP and APC anti-mouse OX40 for two hours. FO-sfGFP, green; APC-anti mouse Flt3 (APC anti-mouse OX40), red. White scale bars, Left, 50 μm ; Right, 10 μm . For **b**, **d-g**, data were mean \pm s.e.m. statistical significance was determined by two-tailed unpaired Student's t-tests. ns represented $p > 0.05$.

Comment 3:

3. The action mechanism of ligands that acts with the receptor of immune cells and causes immune response has not been verified. In the manuscript, authors showed internalization of FOLactis by DC (Fig. 2a), but did not demonstrate downstream signal pathway.

Answer 3:

Thanks for pointing out this for us. Our results showed that FOLactis had a powerful

activating effect on DCs activation. Entry of Lactis into the body elicits a response from host innate immune cells, including DCs. The beneficial effects of LAB on immunity are primarily associated with the specific activation of pattern-recognition receptors (PRRs), especially toll-like receptors (TLR), on the surface of dendritic cells. These receptors recognize LAB-specific molecular patterns, the so-called microbial-associated molecular patterns (MAMPs), which leads to the activation of the multiple cascades of intracellular signaling, especially NF- κ B transcription factor, with the resultant synthesis of endogenous antibacterial factors, defensins, proinflammatory cytokines and chemokines^{10, 11}.

To demonstrate downstream signal pathway, we design the experiment as follows: DCs are stimulated with normal saline, Lactis and FOLactis following the method described in the manuscript. Then the stimulated DCs will be detected by western blot. Changes in the expression of important proteins are shown in the Fig.2c. We observed FOLactis induced a higher expression of TLR1, TLR2 and TLR6 in BMDCs than that in the NS group, which further activated NF- κ B signaling pathway. Next, we studied the immune response stimulated by DC maturation by measuring the secretion of pro-inflammatory cytokines in the culture medium. Compared to the NS, the secretion of TNF- α , IL-6, and IL-1 β increased significantly after treatment of BMDCs with Lactis or FOLactis for 24 hours in vitro (added as Fig. 2d). We also confirmed the activity of the fusion protein FO-sfGFP through their reaction with DCs or T cells. Consistent with the results that FOLactis has a powerful activating effect on DCs and T cells activation, the immunofluorescence co-localization analysis showed that FO-sfGFP could bind Flt3 on DCs and OX40 on T cells (added as Fig. 2h-i).

Comment 4:

4. In Fig. 3, authors demonstrated biodistribution of FOLactis using DiO labeled

bacteria until 360 h after i.t. injection. FOLactis will overgrow in tumour and other tissue when they colonize. DiO signal will weaken as bacteria divide. How can you quantitatively estimate the amount of bacteria in many tissues using this method. I suggest the enumeration of FOLactis counts in different organs at different time points after injection.

Answer 4:

Thanks so much for your professional and careful comments. We completely acknowledge that our results will be more solid and convincing by analyzing the absolute number of FOLactis counts in different organs at different time points after injection. Thus, based on your advice, the major organs, blood and tumour tissue were collected, homogenized, serially diluted, and plated on GM17 plates. By counting the CFU in each plate, it was found that bacterial growth remained restricted to treated tumours and no bacteria could be cultured from untreated tumours or blood and other major organs of treated mice (added as Supplementary Fig.10). The results were consistent with the previous non-quantitative results (Fig. 3).

Supplementary Fig.10 | Bacterial colonization in CT26-bearing mice after intratumoural injection. a-b, Representative photographs of solid GM17 agar plates (a) and quantification

(b) of bacterial colonization in various organs harvested from CT26-bearing mice at different time points after injection of bacteria.

Comment 5:

5. Authors determined the concentration of FO 50 µg in vivo based on the antitumour efficacy study in Supplementary Fig. 6. The rationale of comparing FOLactis 10⁹ CFU with FO 50µg + Lactis 10⁹ CFU is not understandable (Fig. 4b). How much amount of FO can be produced from 10⁹ CFU FOLactis? If it is over 50 µg, why don't you compare with 100 µg FO + 10⁹ CFU Lactis? If therapeutic outcome of 100 µg FO + 10⁹ CFU Lactis is better than that of 10⁹ CFU FOLactis, what is the merit of 10⁹ CFU FOLactis?

Answer 5:

We sincerely apologize for the unclear explanation we made. Previous studies using protein expression systems such as the NICE system have reported that recombinant proteins are expressed primarily during the exponential growth phase^{12, 13}. Thus, we added nisin into FOLactis for protein expression during the exponential growth phase. After three hours of incubation, we collected FOLactis and washed them with PBS for three times. Then we obtained 10⁹ CFU FOLactis by means of agar plate continuously dilution. When the number of FOLactis reached 10⁹ CFU, the absorbance at 600 nm measured by spectrophotometer was about 1.0. As shown in the revised Fig.1c, the amount of FO can be produced from 10⁹ CFU FOLactis is far below 50µg. The 10⁹ CFU FOLactis group obtained the best curative effect for the following reason: FOLactis released FO continuously while FO metabolized so quickly in vivo that there was not enough time for them to exert antitumour effects. Therefore, the FOLactis group has the best tumour inhibitory effect (Fig. 4b-c).

Comment 6:

6. Authors suggested ISV effect of FOLactis based on the data in Fig. 4c. I am not sure that this can be suggested as ISV effect because authors didn't check the presence

of FOLactis in the untreated tumour (opposite side). Tumour reduction in untreated tumour can be interpreted as moving from the location of treated tumour to the opposite site. Authors should count the bacterial number and evaluate the immunological phenotypes in the untreated tumours.

Answer 6:

We appreciate it a lot for your valuable and constructive suggestions. We totally agree that it is significant to exclude the possibility of bacterial trafficking to the untreated tumour and evaluate the immunological phenotypes in the untreated site. As previously reported in the literature^{14, 15}, after unilateral intratumoural injection of FOLactis, untreated tumours were collected, homogenized, serially diluted, and plated on GM17 plates. By counting the CFU in each plate, it was found that bacterial growth remained restricted to treated tumours and no bacteria could be cultured from untreated tumours (added as Supplementary Fig.10). The results were consistent with the previous non-quantitative results. Then we analyzed the activation of T cells in the untreated tumours. Compared to the NS group, the proportion of T_{EM} (CD3⁺CD8⁺CD44⁺CD62L⁻) subset and IFN- γ ⁺ CD8⁺ T cells increased at the untreated site 7 days later in the FOLactis group (added in Supplementary Fig.12j-1).

Supplementary Fig.12 | In situ vaccination with FOLactis modulated the immune microenvironment in the local tumour and TDLN. **j**, Flow cytometric analysis of effector memory T cells (T_{EM}, CD3⁺CD8⁺CD44⁺CD62L⁻) and central memory T cells (T_{CM}, CD3⁺CD8⁺CD44⁺CD62L⁺) in the treated tumour (48h after treatment) and non-treated tumour (168h after treatment). **k-i**, Frequency of tumour infiltrating IFN-γ⁺ within CD8⁺ T cells (168h after treatment). For the experiments in **a-d**, **f-k**, data were mean ± s.e.m. statistical significance was determined by analysis of two-tailed unpaired Student's t-tests. ns represented $p>0.05$.

Comment 7:

7. Authors did not assess the antitumour effect of Lactis expressing Flt3L only or OX40L only with that of FOLactis.

Answer 7:

Thank you very much for the constructive comment. According to your suggestion, we designed the engineered Lactis expressing Flt3L only or OX40L only, and evaluated the antitumour effect in the subcutaneous CT26 tumour model. The results were shown in new Supplementary Fig.8, and were added into the manuscript as follows:

Similar to the construction of FOLactis, we designed the engineered Lactis expressing

Flt3L (FLactis) or OX40L (OLactis) (Supplementary Fig. 8a-b). The successful expression of the protein was confirmed by WB and ELISA (Supplementary Fig. 8c). It was found that the mice in the FOlactis group exhibited a stronger inhibition of tumour growth compared to the mice in other groups, with a marked extension of animal survival over the other groups (Supplementary Fig. 8d-e).

Supplementary Fig.8 | Construction of engineered *Lactococcus lactis* delivering Flt3L

(FLactis) or OX40L (OLactis). **a-b**, plasmid map of a *Lactococcus lactis*-*Escherichia coli* shuttle vector using pNZ8148 expressing Flt3L (FLactis) or OX40L (OLactis). **c**, Western blotting analysis of the induced or non-induced engineered *Lactococcus lactis*. Nisin is an inducer of protein expression. M: molecular mass marker; Lane 0-4: the whole bacteria lysates (bacteria) of wild-type *Lactococcus lactis* (Lactis) induced by nisin, OLactis induced by nisin, non-induced OLactis, FLactis induced by nisin and non-induced FLactis, respectively. Bacteria (10^9 CFU) were collected and the pellets were sonicated. The amounts of the target protein in the bacterial lysates of FLactis / OLactis were assessed by ELISA. **d**, Average tumour-growth curves of BALB/c mice bearing CT26 colon tumour with different treatments as indicated ($n=6$). The mice were administered with NS, 10^9 CFU Lactis, 10^9 CFU FLactis, 10^9 CFU OLactis or 10^9 CFU FOLactis intratumourally on days 7, 9 and 11, which were dissolved in normal saline to a final volume of 100 μ l per dose. The tumour size was measured every 2-3 days from the first administration day. The error bars represented mean \pm s.e.m. p -values were calculated by two-way ANOVA and Tukey post-test and correction. ns represented $p>0.05$. **e**, Survival curves of BALB/c mice in different groups for 60 days ($n=6$). p -values were calculated by log-rank (Mantel-Cox) test. ns represented $p>0.05$.

Comment 8:

8. RNA-seq analysis of tumours after FOLactis treatment, authors observed activation in both innate and adaptive immunity through T cell activation and NK cell activation. I am so curious of the effect of Lactis itself whether Lactis can stimulate innate immunity or not.

Answer 8:

We thank you very much for the important point you have proposed. The host-microbial interactions play an important part in the stimulation of innate immunity. For example, Pattern-recognition receptors (PRRs) in the innate immune system can sense microorganisms through conserved molecular structures. Several families of PRRs and their signalling pathways are now known, including the Toll-like receptors (TLRs), the nucleotide-binding oligomerization (NOD)-like receptors (NLRs), the

RIG-I-like receptors, the C-type lectin receptors, the absent in melanoma 2 (AIM2)-like receptors and the OAS-like receptors¹. These sensors are expressed by a variety of cellular compartments and constitute a continuous surveillance system for the presence of microorganisms in tissues¹⁶. Recently, Katarzyna Garbacz published a review, in which they reported the anticancer activity of lactic acid bacteria (LAB)¹¹. In the study, we know that there are several compounds of LAB that are associated with the stimulation of innate immunity, including LAB peptidoglycan, exopolysaccharides, short-chain fatty acids and so on. Peptidoglycan not only protects the bacterial cell integrity as well as cell shape but also modulates immune responses, stimulates the production of tumour necrosis factors, interferons, and interleukins (IL-1, IL-6, IL-8, IL-12)^{17,18}. Anticancer activity of LAB polysaccharides is associated with the stimulation of immune cells, primarily lymphocytes T and B, macrophages and NK cells, releasing interleukins¹⁹.

According to these studies, we tested the stimulation of innate immunity by incubating BMDCs with bacteria for 24 hours in vitro (added in Fig. 2). Then the stimulated DCs will be detected by western blot and flow cytometry. We observed Lactis also induced a higher expression of TLR1, TLR2 and TLR6 in BMDCs than that in the NS group, which further activated NF- κ B signaling pathway (added in Fig. 2c). Meanwhile, the secretion of TNF- α , IL-6 and IL-1 β also significantly increased in the Lactis group compared to the other groups, which played crucial roles in the initiation and stimulation of innate immune response (added in Fig. 2d). In CT26 mouse colon tumour models, we removed intratumourally treated tumours after two days from the last treatment to detect the changes of immune cells. There was a significantly higher proportion of NK cells in Lactis-containing groups (Lactis, Lactis + FO, FOLactis), suggesting Lactis contributed mostly to the increase of NK cells (Fig. 5f). We hope our revisions can address your concern.

Comment 9:

9. Supplementary Fig 10 and 11 data showed biosafety of FOLactis on 7 dpi. Toxicity

such as cytokine storm can occur earlier than 7 dpi. Serum study on 3 dpi would be needed.

Answer 9:

Thank you for your great concern about the safety of FOLactis in vivo. We fully recognize that life-threatening side effects, such as a cytokine storm, may occur with the enhanced stimulation of the patient immune system during cancer immunotherapy, especially along with the existence of bacteria as drugs. Considering the biosafety of FOLactis, we measured the concentrations of inflammatory cytokines and chemokines in the serum two days and seven days after the last treatment, evaluating the response of systemic inflammation stimulated by the FOLactis. Overall, the serum concentrations of IFN γ , IL-2, IL-13, IL-4, IL-10, IL-5, IL-6, and TNF- α were generally similar among all the groups without a statistical difference, indicating that ISV with FOLactis had a good biological safety (Supplementary Fig. 17a, Supplementary Fig. 18a).

Supplementary Fig.18 | FOLactis induced low systemic inflammation. a, The levels of interferon-gamma (IFN γ), IL-2, IL-13, IL-4, IL-10, IL-5, IL-6, and TNF- α in serum from CT26 tumour-bearing mice isolated two days after the last treatment (n=6). Data were mean \pm s.e.m. p-values were determined by two-tailed unpaired Student's t-tests. ns represented p>0.05.

Comment 10:

10. The first paragraph of Discussion should move to the initial part of Introduction.

Answer 10:

Thank you for the helpful comment. We have moved it to the initial part of introduction in the manuscript.

Reviewer #2 (Remarks to the Author): with expertise in bacteria-based cancer therapy

This the first submission of this manuscript, Zhu et. al., reporting that they developed a probiotic bacterium *Lactococcus lactis* expressing a fusion protein consisting of Fms-like tyrosine kinase 3 ligand and co-stimulator OX40 ligand. This recombinant *L. lactis* was further evaluated as in situ vaccine. The vaccination regimen showed significant efficacy in three different tumour models, key components of the anti-tumour immune response were activated such as NK cells and CD8⁺. When combined with immune check point inhibitor anti-PD1 the anti-tumour efficacy was greater than each agent separately. This is a novel study with a great potential to influence the field of probiotic immunotherapy. However, there are some weaknesses noted as follow:

Comment 1:

1. The authors claim that the fusion protein FO is sustainelly released. However, the construction did not include a secretion signal, how the fusion protein is released?

Answer 1:

We are very grateful for your kind suggestions, and fully agree that it is of great value to evaluate whether the fusion protein can be well secreted out of the engineered *lactis* and retain its biological structure and activity. There are some other studies¹⁻³ using bacteria that express proteins intracellularly as potential therapeutics, which proves the feasibility of this strategy. When bacteria invade the body, they are attacked by immune system via three main ways: (i) via complement-mediated lysis^{4, 5}; (ii) via phagocytosis⁶⁻⁸; (iii) via cell-mediated immunity⁹.

According to these studies, we assume that the fusion protein will release as bacteria die. We constructed an engineered *Lactis* (FOLactis-sfGFP) expressing the fusion protein Flt3L-OX40L-sfGFP (added in Supplementary Fig.4). After the induction of nisin, FOLactis-sfGFP turned green, indicating that the expressed heterologous protein retained its biological structure and biological activity, which is an important characteristic for successful immunoregulation. Besides, we detected the release of the fusion proteins. FOLactis-sfGFP released the fusion protein in PBS slowly and

continuously, while it took lesser time to release the fusion protein in PBS mixed with BMDCs or splenocytes (added in Supplementary Fig. 4c, e-f). The corresponding results have been revised in Supplementary Fig.4.

Supplementary Fig.4 | Sustained release of the fusion protein from the engineered

Lactis. **a**, plasmid map of a *Lactococcus lactis*-*Escherichia coli* shuttle vector using pNZ8148 expressing the Flt3L-OX40L-sfGFP fusion protein. **b**, Cellular uptake of FOLactis-sfGFP after a two-hour incubation with BMDCs, as assessed by flow cytometry (n=3, biologically independent samples). **c**, 10^9 CFU FOLactis or FOLactis-sfGFP were resuspended in PBS at 37 ° C. In vitro images to demonstrate the activity and release of the Flt3L-OX40L-sfGFP fusion protein. **d**, Colocalization analysis of FOLactis-sfGFP (green) with lysosomal compartments (red) in BMDCs by confocal microscopy (two-hour incubation). White scale bars, 10 μ m. **e-f**, Curves of the Flt3L-OX40L-sfGFP fusion protein release from the engineered Lactis with or without other cells in PBS at 37 ° C.

Comment 2:

2. The author also claim that *L. lactis* expressing FO fusion protein could kill cancer cells by direct cell lysis. However, there is no data or experiments demonstrating such effect.

Answer 2:

Thanks so much for your professional and careful comments. Recently, Katarzyna Garbacz published a review, in which they reported the anticancer activity of lactic acid bacteria (LAB)¹¹. In the study, we know that there are several compounds of LAB that are associated with the enhanced apoptosis of cancer cells, including LAB bacteriocins, peptidoglycan, surface layer, exopolysaccharides and so on. For example, peptidoglycan in the cell wall fraction of LAB strains was reported to have cytotoxic activities in some tumour cell lines in vitro and had a significantly antiproliferative effect on different cancer cell lines²⁰. Similarly, Wang et al. found that whole peptidoglycan (WPG) from the *Lactobacillus paracasei* subsp. *paracasei* M5 strain inhibits the proliferation of colon cancer HT-29 cells and induces apoptosis. The apoptosis property of peptidoglycan was mediated by upregulating proapoptotic genes, downregulating antiapoptotic genes and promoting the release of cytochrome C in the mitochondria to the cytosol²¹. LAB exopolysaccharides (EPS) were shown to exert an antiproliferative effect in various cancer cell lines in a dose- and time-dependent

manner. The most extensively studied cell lines originated from intestinal, liver and breast malignancies²². According to many authors, LAB preparations exerted the strongest antiproliferative effect in intestinal cancer cell lines, which implies that their therapeutic potential in the case of these malignancies might be the highest²³⁻²⁵. The primary mechanism of LAB-induced apoptosis is known to be based on the activation of extrinsic and intrinsic pathways controlled by caspases, the proteases specific for cysteinyl aspartate²⁶.

As previously described²⁷, we incubated tumour cells (CT26-Luc, 4T1-Luc, MC38-GFP) with FOLactis and/or splenocytes in vitro for 24 hours. Then we tested the luciferase activity or fluorescence of GFP to evaluate tumour cell killing. The results showed that FOLactis could kill cancer cells by direct cell lysis and the tumour killing rate was the highest in the FOLactis + splenocytes group. We also observed the morphological changes of cancer cells after treatment for 24 hours. These cancer cells produced large quantities of balloon-shaped vesicles (added in Fig. 2g, Supplementary Fig. 7a). Cell death induction was further confirmed by increased staining with Propidium Iodide (PI), a cell impermeable dye that stains cells that have lost membrane integrity (added in Supplementary Fig. 7b). We hope our explanations and revisions can answer your concerns.

Fig.2 | FOLactis efficiently enhanced the activation of DCs and T cells, and induced tumour cell death in vitro. **g**, Luminescence was detected by the addition of D-luciferin to CT26-luc cells after incubated with T cells and/or FOLactis for 24 hours (n=6). Cells were examined under the microscope EVOS FL Auto Cell Imaging System (Invitrogen) and photographed. White scale bars, 200 μ m. data were mean \pm s.e.m. statistical significance was determined by two-tailed unpaired Student's t-tests. ns represented $p>0.05$.

Supplementary Fig.7 | FOlactis could kill cancer cells by direct cell lysis. a, Luminescence was detected by the addition of D-luciferin to 4T1-Luc cells after incubated with T cells and/or FOlactis for 24 hours (n=6). Cells were examined under the microscope EVOS FL Auto Cell Imaging System (Invitrogen) and photographed. White scale bars, 200 μ m. **b,** Fluorescence was detected after MC38-GFP cells incubated with T cells and/or FOlactis for 24 hours (n=6). Cells were examined under a confocal laser scanning microscopy (Leica, Germany). White scale bars, 50 μ m. data were mean \pm s.e.m. statistical significance was determined by two-tailed unpaired Student's t-tests. ns represented $p>0.05$.

Comment 3:

3. The authors hypothesize that *Lactococcus lactis* could increase the expression of OX40 on CD4⁺ T cells in TME by PAMPs. However, the reference 20 is not about pathogen-associated molecular patterns. It is not clear whether this is case because *L. lactis* lacks of lack of toxicity and immunogenicity. <https://www.frontiersin.org/articles/10.3389/fimmu.2017.01961/full>

Answer 3:

We apologize for any misunderstandings caused by not properly labeling and presenting the literature. We have revisited the relevant citations and stated as the following: PAMPs on the surface of bacteria can be accessed by PRRs on the surface of DCs, which transmit relevant signals to T cells to enable the immune system to function. Among PRRs, TLRs are the most important category, and correspondingly, PAMPs contain a variety of TLRs agonists, including LPS (TLR4 agonist)²⁸. The work of citation 20 of the previous manuscript confirmed that CpG, as an agonist of TLR9, can increase OX40 on the surface of T cells in vivo, and myeloid cells such as DCs and macrophages are indispensable in this process²⁹. Therefore, we hypothesized that *Lactococcus lactis* can activate DCs through PAMPs including TLRs on its surface, and DCs transmit relevant signals to T cells, thereby upregulating OX40 on the surface of T cells. The work of citation 20 is very important for us to assume the above conclusion, but this work is indeed not discussed from the perspective of PAMPs; in addition, for the literature you mentioned³⁰, it demonstrated that lactic acid bacteria are a safe drug carrier for humans, as exogenous bacteria enter the human body. It will cause a mild non-specific immune response, but will not produce obvious toxicity, which is also an important factor for us to choose lactic acid bacteria as a non-specific immune adjuvant.

Meanwhile, we do observe that *Lactis* induce a higher expression of TLR1, TLR2 and TLR6 in BMDCs than that in the NS group, which might further increase the expression of OX40 on CD4⁺ T cells. Thank you for your detailed comments that have helped us improve the manuscript. To avoid similar oversights, we checked all the content of the manuscript and revised inappropriate descriptions. These places were highlighted in yellow in the revised manuscript.

Fig.2 | FOLactis efficiently enhanced the activation of DCs and T cells, and induced tumour cell death in vitro. c, Western blot analysis of several membrane TLR proteins and NF-κB pathway in BMDCs. The gels were loaded with equal amounts of the proteins (10 µg).

Supplementary Fig.12 | In situ vaccination with FOLactis modulated the immune microenvironment in the local tumour and TDLN. e-f, When intratumoural injection either with normal saline (NS) or Lactis once, tumours were excised 48 hours later, and OX40 expression of the CD3⁺CD4⁺ T cell subset was analyzed by flow cytometry (n = 6).

Comment 4:

4. The authors mentioned that they used orthotopic mouse models but in reality they used subcutaneous tumour models.

Answer 4:

Thank you for pointing out this mistake. Animals used in this study were all subcutaneous tumour models. Moreover, we repeated our in vivo anti-tumour study in the orthotopic 4T1 model. We confirmed and substantiated our conclusion that FOLactis could significantly inhibit tumour growth, reduce lung metastasis and increase the survival rate of tumour-bearing mice (added in Supplementary Fig.15a-e). We have revised the description and checked the whole manuscript.

Supplementary Fig.15 | FOLactis inhibited tumour growth and metastasis in the orthotopic 4T1 tumours. a-c, Orthotopic 4T1-bearing mice received treatments on days 7, 9, and 11 (n=6). Anterior bioluminescence images of tumour burden on days 7, 14, 21, and 28 after tumour inoculation (a). Shown are the tumour signal (b) and overall survival (c). d, Representative IVIS images of lungs extracted from 4T1-Luciferase orthotopic tumours. e, Quantification of metastatic lesions in lung tissues. For the experiments in b and e, data were the mean \pm s.e.m. *p*-values were determined by two-tailed unpaired Student's *t*-tests. ns represented *p*>0.05. For the experiments in c, *p*-values were calculated by log-rank (Mantel-Cox) test. ns represented *p*>0.05.

Comment 5:

5. In the line 248 the authors used the word "obviously" this word is not acceptable in a scientific manuscript.

Answer 5:

Thank you very much for your corrections. We have replaced "obviously" with "significantly" in describing the comparison of tumour inhibition.

Reviewer #3 (Remarks to the Author): with expertise in cancer immunology

In this work, the authors describe a new immunotherapy strategy based on the use of a bacterial vector harboring a chimeric protein which has two immunomodulatory domains: FLt3L and OX40L. The results presented are consistent, and very interesting, demonstrating the potential of using the strategy for the development innovative clinical approaches. I have a few questions to discuss and also minor observations, that may contribute to the paper.

Questions to discuss:

Comment 1:

1. Flt3L has been described to suppress GVHD associated to DCs signaling, inhibiting T cells (<https://doi.org/10.3389/fimmu.2021.699128>). On the other hand, Flt3L enhances DC expansion and activity that may contribute to anti-cancer therapies. Please discuss about switching this balance between immunotolerance and enhancing antitumour activity. How can you drive Flt3L signaling to antitumour response?

Answer 1:

Thank you for your comments that sparked our thinking about the role of Flt3L in different diseases, and we have read this literature you mentioned in detail. GVHD refers to the main complication of allogeneic hematopoietic stem cell transplantation, which is caused by T lymphocytes in the allogeneic donor graft after transplantation, targeting the recipient's target organ to launch a cytotoxic attack. As mentioned in the literature that "Flt3L has been found to suppress graft-versus-host disease (GvHD), specifically via host DCs", Flt3L promotes the host's immune function while suppressing the attack of allogeneic donor T cells on the host³¹. That is, Flt3L can promote immunotolerance against GVHD caused by allogeneic T cells through host DCs, as well as promote the host's own anti-cancer therapies³². From this point of view, 'this balance between immunotolerance and enhancing antitumour activity' is actually derived from the two manifestations of Flt3L's immune-promoting effect

rather than opposition. Since our experiment was only concerned with enhancing autoimmunity to exert anti-tumour effects, we focused on the promoting effect of Flt3L on DCs. We hope our explanation can address your concerns.

Comment 2:

2. Please comment about figure 4h-j, whether depletion of CD4 and CD8 cells includes both DCs and T cells. I'm curious if authors also tested anti-CD3 depletion.

Answer 2:

Thank you for your constructive questions. Regarding whether depletion of CD4 and CD8 cells includes both DCs and T cells, we have thoroughly consulted the introduction of antibody on the website and some related literatures. The introduction demonstrated their antibodies as follows: (i) CD8 is primarily expressed on the surface of cytotoxic T cells, but can also be found on thymocytes, natural killer cells, and some dendritic cell subsets. Its reported applications include in vivo CD8⁺ T cell depletion. <https://bxcell.com/product/invivoplus-anti-m-lyt-2-2-cd8a/> (ii) CD4 is expressed by the majority of thymocytes, most helper T cells, a subset of NK-T cells and weakly by dendritic cells and macrophages. CD4 plays an important role in the development of T cells and is required for mature T cells to function optimally. Its reported applications include in vivo CD4⁺ T cell depletion. <https://bxcell.com/product/invivoplus-anti-m-cd4/> In addition, the introduction of CD3 antibody shows that CD3 ϵ is expressed on T lymphocytes, NK-T cells, and to varying degrees on developing thymocytes and its application includes in vivo T cell depletion. <https://bxcell.com/product/m-cd3e-fab2-fragments/> To conclude, depletion of CD4 and CD8 cells includes mainly T cells and few other cells.

Similar to other published literatures^{29,33,34}, we focused on the function of specific T cell subsets. Naive T cells differentiate into several functional classes of effector T cells that are specialized for different activities on recognizing antigen. CD8 T cells recognize pathogen peptides presented by MHC class I molecules, and naive CD8 T

cells differentiate into cytotoxic effector T cells that recognize and kill infected cells. CD4 T cells can differentiate down distinct pathways that generate effector subsets with different immunological functions after recognizing pathogen peptides presented by MHC class II molecules. Although CD8 T cells and CD4 T cells belong to T cells, they play different roles in cancer immunotherapy. In our study, ISV with FOLactis could induce tumour regression mainly dependent on CD8 T cells, with CD4 T cells playing lesser roles in tumour rejection.

Comment 3:

3. Authors propose to use bacterial vectors expressing FO as a drug for cancer therapy. Since bacterial administration into sterile body compartments (as tumour site) may elicit immune response, it was shown several experiments to investigate adverse. These experiments were performed challenging animals with bacterial vector. I think it should be interesting to rechallenge animals and evaluating adverse effects. Another related question to discuss: what happens if these bacterial vectors enter blood stream?

Answer 3:

Thanks so much for your kind suggestions. We have repeated the tumour suppression experiment in CT26 tumour-bearing mice with FOLactis. FOLactis-treated and cured mice were rechallenged with 10^9 CFU FOLactis subcutaneously at the same site of primary tumour for three times after two months. Then, we performed hematological examinations, including serum biochemistry assays, two days after the treatment. As shown in the **Answer Fig. 1**, there was no significant difference in the serum biochemistry indexes of the mice in the NS and FOLactis group, and all of them were within the normal range.

We analyzed the absolute number of FOLactis counts in different organs at different time points after injection. Blood was collected and plated on GM17 plates. By counting the CFU in the plate, it was found that no bacteria could be cultured from blood (added as Supplementary Fig.10). Thus, we concluded FOLactis did not exist in

blood six hours after the last intratumoural injection. In addition, male BALB/c mice bearing subcutaneous CT26 tumours were intravenously injected with FOLactis at the dose of 10^9 CFU per mouse once and then sacrificed at 48 hours after injection. We collected blood and plate it on GM17 plates. The results showed that there was no bacteria in the blood (Answer Fig. 2). We speculated that FOLactis were quickly removed from the bloodstream.

Answer Fig. 1 | Biosafety assessment of FOLactis when rechallenging animals.

FOLactis-treated and cured mice were rechallenged with 10^9 CFU FOLactis subcutaneously at the same site of primary tumour for three times after two months. Blood biochemistry and hematology data of male BALB/c mice two days after the last treatment (n=3). ALT, glutamic-pyruvic transaminase; AST, aspartate aminotransferase; Statistic was based on three mice per data point. Data were mean \pm s.e.m. p-values were determined by two-tailed unpaired Student's t-tests. ns represented $p > 0.05$.

Supplementary Fig.10 | Bacterial colonization in CT26-bearing mice after intratumoural injection. a-b, Representative photographs of solid GM17 agar plates (a) and quantification

(b) of bacterial colonization in various organs harvested from CT26-bearing mice at different time points after injection of bacteria.

Answer Fig.2 | Bacterial colonization in blood in CT26-bearing mice after intravenous injection. Representative photographs of GM17 plates of bacterial colonization in blood harvested from CT26-bearing mice 48 hours after intravenous injection of FOLactis.

Comment 4:

4. When authors purified the FO, it was described a complex protocol for protein purification from inclusion bodies, that are expected as Flt3L and OX40L are eukaryotic membrane proteins. If these proteins should be purified like that, why does bacterial vector expressing FO secretes these recombinant protein? Please clarify.

Answer 4:

We appreciate it a lot for your professional and detailed comments regarding the production and secretion of FO in FOLactis. *Escherichia coli* is widely used as an expression system for production of recombinant proteins of prokaryotic and eukaryotic origin. However, the biggest obstacle lies in obtaining large amounts of a given protein in a correctly folded form. When a large number of proteins are expressed, *E. coli* is prone to produce inclusion bodies because *E. coli* has two membranes and proteins are easy going to a periplasmic space, where they do not have enough time to fold correctly^{35,36}. In addition, the *E. coli* strain has wildtype LPS expression and LPS can be fatal in excess of a certain amount³⁵. Thus, though we

obtain FO from E.coli, we need to refold the unfolded recombinant protein and remove the endotoxin, which is a complex process.

Compared to E. coli, Lactococcus lactis has the advantage of being GRAS (Generally Recognized As Safe) and endotoxin free, important for both therapeutic and food applications. NICE (nisin-inducible controlled gene expression) is the most widely used inducible system in this microorganism, which affords tightly controlled expression and relatively high protein expression. Moreover, Lactococcal system is advantageous in having only one membrane, which potentially makes membrane-targeting and processing easier and less prone to inclusion body formation³⁵. Since they have an excellent safety profile and have antitumour activity on their own¹¹, we can use both Lactis and their produced protein as anticancer agents so that it is not necessary to purify protein from engineered Lactis, which is convenient and efficient.

When bacteria invade the body, they are attacked by immune system via three main ways: (i) via complement-mediated lysis^{4, 5}; (ii) via phagocytosis⁶⁻⁸; (iii) via cell-mediated immunity⁹. There are some other studies¹⁻³ using bacteria that express proteins intracellularly as potential therapeutics, which proves the feasibility of this strategy.

According to these studies, we assume that the fusion protein will release as bacteria die. We constructed an engineered Lactis (FOLactis-sfGFP) expressing the fusion protein Flt3L-OX40L-sfGFP (added in Supplementary Fig.4). After the induction of nisin, FOLactis-sfGFP turned green, indicating that the expressed heterologous protein retained its biological structure and biological activity, which is an important characteristic for successful immunoregulation. Besides, we detected the release of the fusion proteins. The corresponding results have been revised in Supplementary Fig.4.

Comment 5:

5. Technical question 1: line 100: why it is important to have “extracellular domain of the Flt3L at the N-terminus fused to the extracellular domain of the OX40L at the C-terminus by a matrix metalloproteinase (MMP)-sensitive peptide Pro-Val-Gly-Leu-Iso-Gly ? Is it important to separate flt3L from OX40L for biological effect? I think you used same strategy for “FOLactis” (supp fig 1, supp fig 2)? Have you compared the lactis + flt3L + OX40L with lactis expressing a fusion protein flt3L-OX40L that hasn't MMP site? Is the fusion protein harboring MMP active only in the tumour site?

Answer 5:

Thanks a lot for your in-depth comments. Matrix metalloproteinases (MMPs) are highly concentrated only in tumours³⁷. In the initial design, we used MMP-sensitive peptide Pro-Val-Gly-Leu-Iso-Gly to produce FO and FOLactis, trying to make the most of Flt3L and OX40L in the tumour microenvironment respectively. Meanwhile, this strategy improves the possibility of safe and targeted delivery. The fusion protein can only be cleaved by MMP and play their roles in the local tumour, minimizing the effects on the other tissues of the body even with systemic injection.

In our study, we found that both FO and FOLactis could act as immune stimulatory agents on DCs and T cells activation in vitro (added as Fig. 2b, e-f), which meant that the fusion protein was still active to some extent without MMP. Since we used the intratumoural injection as administration route in the study and it was found that bacterial growth remained restricted to treated tumours and no bacteria could be cultured from untreated tumours or blood and other major organs of treated mice (added as Supplementary Fig.10), the MMP-sensitive peptide Pro-Val-Gly-Leu-Iso-Gly might be not that important in this administration route. In a conclusion, MMP-sensitive peptide Pro-Val-Gly-Leu-Iso-Gly has the potential to ensure the safety of FOLactis and maximize the effect of the fusion protein so that we design the special peptide in the first. We hope our revisions can answer your concern.

Comment 6:

6. Technical question 2: the FO protein expressed in E coli should be purified from inclusion bodies, on the other hand, authors mention that *Lactococcus lactis* may be engineered to produce a FO protein that is secreted, enhancing immunity. What is different in FO expression in *lactis* that FO protein doesn't go to inclusion bodies and is secreted?

Answer 6:

Thanks for raising this concern for us. Inclusion bodies are dense, insoluble protein particles wrapped by membranes formed when exogenous genes are expressed in prokaryotic cells, especially in *Escherichia coli* (*E. coli*). *E. coli* and *Lactococcus lactis* are two prokaryotic systems commonly used for membrane protein production. *E. coli* has an inner and outer membrane, which leads to proteins easily going to a periplasmic space. Overexpression of recombinant proteins may lead to the accumulation of insoluble protein aggregates and the formation of inclusion bodies. In comparison, Lactococcal system is advantageous in having only one membrane, which potentially makes membrane-targeting and processing easier and less prone to inclusion body formation.

In our study, we constructed an engineered *Lactis* (FO*Lactis*-sfGFP) expressing the fusion protein Flt3L-OX40L-sfGFP (added in Supplementary Fig.4). After the induction of nisin, FO*Lactis*-sfGFP turned green, indicating that the expressed heterologous protein retained its biological structure and biological activity, which is an important characteristic for successful immunoregulation.

Supplementary Fig.4 | Sustained release of the fusion protein from the engineered Lactis. **a**, plasmid map of a *Lactococcus lactis*-*Escherichia coli* shuttle vector using pNZ8148 expressing the Fit3L-OX40L-sfGFP fusion protein. **b**, Cellular uptake of FOLactis-sfGFP after a two-hour incubation with BMDCs, as assessed by flow cytometry (n=3, biologically independent samples). **c**, 10^9 CFU FOLactis or FOLactis-sfGFP were resuspended in PBS at 37°C . In vitro images to demonstrate the activity and release of the Fit3L-OX40L-sfGFP

fusion protein. **d**, Colocalization analysis of FOLactis-sfGFP (green) with lysosomal compartments (red) in BMDCs by confocal microscopy (two-hour incubation). White scale bars, 10 μ m. **e-f**, Curves of the Flt3L-OX40L-sfGFP fusion protein release from the engineered Lactis with or without other cells in PBS at 37 ° C.

Comment 7:

We sincerely appreciate your careful review and revised the manuscript.

(1) Abstract: I suggest reviewing English language. Egg Lines 28-29 a lot of “and”

Answer (1):

We have revised the description with fewer “and” in the abstract.

Intratumoural delivery of FOLactis would contribute to local retention and sustained release of therapeutics to thoroughly modulate key components of the antitumour immune response, such as activation of natural killer (NK) cells, cytotoxic T lymphocytes, and conventional-type-1-dendritic cells in the tumours and tumour-draining lymph nodes.

(2) Line 75 why does Injection of FOLactis could kill cancer cells by direct cell lysis?

I couldn't find this explanation in the indicated references (14-16)

Answer (2):

Thanks so much for your professional and careful comments. Recently, Katarzyna Garbacz published a review, in which they reported the anticancer activity of lactic acid bacteria (LAB)¹¹. In the study, we know that there are several compounds of LAB that are associated with the enhanced apoptosis of cancer cells, including LAB bacteriocins, peptidoglycan, surface layer, exopolysaccharides and so on. For example, peptidoglycan in the cell wall fraction of LAB strains was reported to have cytotoxic activities in some tumour cell lines in vitro and had a significantly antiproliferative effect on different cancer cell lines²⁰. Similarly, Wang et al. found that whole peptidoglycan (WPG) from the *Lactobacillus paracasei* subsp. *paracasei* M5 strain inhibits the proliferation of colon cancer HT-29 cells and induces apoptosis. The

apoptosis property of peptidoglycan was mediated by upregulating proapoptotic genes, downregulating antiapoptotic genes and promoting the release of cytochrome C in the mitochondria to the cytosol²¹. LAB exopolysaccharides (EPS) were shown to exert an antiproliferative effect in various cancer cell lines in a dose- and time-dependent manner. The most extensively studied cell lines originated from intestinal, liver and breast malignancies²². According to many authors, LAB preparations exerted the strongest antiproliferative effect in intestinal cancer cell lines, which implies that their therapeutic potential in the case of these malignancies might be the highest²³⁻²⁵. The primary mechanism of LAB-induced apoptosis is known to be based on the activation of extrinsic and intrinsic pathways controlled by caspases, the proteases specific for cysteinyl aspartate²⁶.

As previously described²⁷, we incubated tumour cells (CT26-Luc, 4T1-Luc, MC38-GFP) with FOLactis and/or splenocytes in vitro for 24 hours. Then we tested the luciferase activity or fluorescence of GFP to evaluate tumour cell killing. The results showed that FOLactis could kill cancer cells by direct cell lysis and the tumour killing rate was the highest in the FOLactis + splenocytes group. We also observed the morphological changes of cancer cells after treatment for 24 hours. These cancer cells produced large quantities of balloon-shaped vesicles (added in Fig. 2g, Supplementary Fig. 7a). Cell death induction was further confirmed by increased staining with Propidium Iodide (PI), a cell impermeable dye that stains cells that have lost membrane integrity (added in Supplementary Fig. 7b). We hope our explanations and revisions can answer your concerns.

Fig.2 | FOLactis efficiently enhanced the activation of DCs and T cells, and induced tumour cell death in vitro. g, Luminescence was detected by the addition of D-luciferin to

CT26-luc cells after incubated with T cells and/or FOLactis for 24 hours (n=6). Cells were examined under the microscope EVOS FL Auto Cell Imaging System (Invitrogen) and photographed. White scale bars, 200 μ m. data were mean \pm s.e.m. statistical significance was determined by two-tailed unpaired Student's t-tests. ns represented $p>0.05$.

Supplementary Fig.7 | FOLactis could kill cancer cells by direct cell lysis. a, Luminescence was detected by the addition of D-luciferin to 4T1-Luc cells after incubated with T cells and/or FOLactis for 24 hours (n=6). Cells were examined under the microscope EVOS FL Auto Cell Imaging System (Invitrogen) and photographed. White scale bars, 200 μ m. **b,** Fluorescence was detected after MC38-GFP cells incubated with T cells and/or FOLactis for 24 hours (n=6). Cells were examined under a confocal laser scanning microscopy (Leica, Germany). White scale bars, 50 μ m. data were mean \pm s.e.m. statistical significance was determined by two-tailed unpaired Student's t-tests. ns represented $p>0.05$.

(3) Line 88 – what’s “multiple orthotopic” models were used?

Answer (3):

Thank you for pointing out this mistake. Animals used in this study were all subcutaneous tumour models. Moreover, we repeated our in vivo anti-tumour study in the orthotopic 4T1 model. We confirmed and substantiated our conclusion that FOLactis could significantly inhibit tumour growth, reduce lung metastasis and increase the survival rate of tumour-bearing mice (added in Supplementary Fig.15a-e). We have revised the description and checked the whole manuscript.

Supplementary Fig.15 | FOLactis inhibited tumour growth and metastasis in the orthotopic 4T1 tumours. a-c, Orthotopic 4T1-bearing mice received treatments on days 7, 9, and 11 (n=6). Anterior bioluminescence images of tumour burden on days 7, 14, 21, and 28 after tumour inoculation (**a**). Shown are the tumour signal (**b**) and overall survival (**c**). **d,** Representative IVIS images of lungs extracted from 4T1-Luciferase orthotopic tumours. **e,** Quantification of metastatic lesions in lung tissues. For the experiments in b and e, data were the mean \pm s.e.m. p -values were determined by two-tailed unpaired Student's t-tests. ns represented $p>0.05$. For the experiments in c, p -values were calculated by log-rank (Mantel-Cox) test. ns represented $p>0.05$.

(4) Figure 1c: Elisa assay isn't clear. Please explain y axis and x axis. What does it means murine flt3L/OX40L (pg/mL-1)? Are you normalizing Fl3tL per OX40L? - in the X axis, what does it means FOLactis:Flt3L?

Answer (4):

We apologize for any misunderstandings caused by not properly labeling.

We have revised the information of result as follows:

Fig.1 | Construction of engineered *Lactococcus lactis* delivering Flt3L-OX40L fusion protein (FOLactis). c, Bacteria (10⁹ CFU, OD≈1 in PBS) were collected and the pellets were sonicated. The amounts of the target protein in the bacterial lysates of FOLactis were assessed by ELISA.

(5) Line 108: “According to the manufacturer's instructions” which manufacture? For BL21? For FO ?

Answer (5):

FO protein was expressed in *Escherichia coli* (*E. coli*) BL21 (DE3) after induction by isopropyl β-D-1-thiogalactopyranoside. The cells were then harvested by centrifugation, and suspended and disrupted by sonication. The supernatant of the cell lysate was refolded and further purified using nickel–nitrilotriacetic acid affinity chromatography under native conditions using an AKTA system HisTrap HP column (GE healthcare, CT, USA). It is for FO.

(6) Fig 2b caption, please describe concentration for FO, Lactis and FOLactis that

correspond to the graph bars (FO: 1-100ug/ml, lactis and FOLactis: 10⁽⁵⁾ to 10⁽⁸⁾ cfu/mL).

Answer (6):

We have revised the description more specific in the manuscript and supporting information.

Fig.2 | FOLactis efficiently enhanced the activation of DCs and T cells, and induced tumour cell death in vitro. b, The percentage of mature DCs (mDCs, CD11c+CD80+CD86+) after co-incubation with Lactis or FOLactis in vitro for 24 hours (n=6).

Supplementary Fig.5 | The Flt3L-OX40L fusion protein (FO) and FOLactis efficiently induced the activation of DCs and T cells in vitro. a, The percentage of mature DCs (mDCs, CD11c+CD80+CD86+) after co-incubation with different concentrations of FO or the crude lysates from Lactis or the crude lysates from FOLactis in vitro for 24 hours (n=3).

(7) Fig 7i: I think it should be written anti-PD1 instead PD1 in graph legend?

Caption of figure 7: there are squares instead characters line 441.

Answer (7):

We have revised the description in the manuscript and checked the whole manuscript.

(8) Suppl. Fig 1: I suggest clarifying to highlight sequences that are shown in “B”, to match with the scheme “a”: Flt3L sequence in orange, PVLIG sequence in blue and OX40 in red. What is pnis? What are sequences labelled in RED in “b”, that you describe as best matched peptides?

Answer (8):

We have revised the description in the supporting information. Pnis means nisin promotor, which is an intrinsic part of pNZ8148. When nisin is added, Pnis will control the production of protein. We performed secondary mass spectrometry on the protein and sequences labelled in RED in “b” means two best matched peptides.

(9) Supplementary fig 10 , caption , line 3: there are squares instead characters.

Answer (9):

We have revised the description in the supporting information.

(10) Supplementary fig 11, I thing it should be interferon gamma (IFN γ) instead “IFN-r” y axis.

Answer (10):

We have revised the description in the supporting information.

1. Jeong, H., Lee, S.Y., Seo, H. & Kim, B.J. Recombinant Mycobacterium smegmatis delivering a fusion protein of human macrophage migration inhibitory factor (MIF) and IL-7 exerts an anticancer effect by inducing an immune response against MIF in a tumor-bearing mouse model. *J Immunother Cancer* **9** (2021).
2. Kim, B.J. et al. Recombinant Mycobacterium smegmatis with a pMyong2 vector expressing Human Immunodeficiency Virus Type I Gag can induce enhanced virus-specific immune responses. *Sci Rep* **7**, 44776 (2017).
3. Namai, F. et al. Construction of Genetically Modified Lactococcus lactis Producing Anti-human-CTLA-4 Single-Chain Fragment Variable. *Mol Biotechnol* **62**, 572-579 (2020).
4. Berends, E.T. et al. Distinct localization of the complement C5b-9 complex on Gram-positive bacteria. *Cell Microbiol* **15**, 1955-1968 (2013).
5. Jeanneau, C. et al. Can Pulp Fibroblasts Kill Cariogenic Bacteria? Role of Complement Activation. *J Dent Res* **94**, 1765-1772 (2015).
6. Gordon, S. Phagocytosis: An Immunobiologic Process. *Immunity* **44**, 463-475

- (2016).
7. Underhill, D.M. & Goodridge, H.S. Information processing during phagocytosis. *Nat Rev Immunol* **12**, 492-502 (2012).
 8. Savina, A. & Amigorena, S. Phagocytosis and antigen presentation in dendritic cells. *Immunol Rev* **219**, 143-156 (2007).
 9. Leung, S. et al. The cytokine milieu in the interplay of pathogenic Th1/Th17 cells and regulatory T cells in autoimmune disease. *Cell Mol Immunol* **7**, 182-189 (2010).
 10. Rakoff-Nahoum, S., Paglino, J., Eslami-Varzaneh, F., Edberg, S. & Medzhitov, R. Recognition of commensal microflora by toll-like receptors is required for intestinal homeostasis. *Cell* **118**, 229-241 (2004).
 11. Garbacz, K. Anticancer activity of lactic acid bacteria. *Semin Cancer Biol* (2022).
 12. Namai, F., Shigemori, S., Ogita, T., Sato, T. & Shimosato, T. Construction of genetically modified *Lactococcus lactis* that produces bioactive anti-interleukin-4 single-chain fragment variable. *Mol Biol Rep* **47**, 7039-7047 (2020).
 13. de Ruyter, P.G., Kuipers, O.P. & de Vos, W.M. Controlled gene expression systems for *Lactococcus lactis* with the food-grade inducer nisin. *Appl Environ Microbiol* **62**, 3662-3667 (1996).
 14. Chowdhury, S. et al. Programmable bacteria induce durable tumor regression and systemic antitumor immunity. *Nat Med* **25**, 1057-1063 (2019).
 15. Yi, X. et al. Bacteria-triggered tumor-specific thrombosis to enable potent photothermal immunotherapy of cancer. *Sci Adv* **6**, eaba3546 (2020).
 16. Thaiss, C.A., Zmora, N., Levy, M. & Elinav, E. The microbiome and innate immunity. *Nature* **535**, 65-74 (2016).
 17. Jafarei, P. & Ebrahimi, M.T. *Lactobacillus acidophilus* cell structure and application. *African Journal of Microbiology Research* **5**, 4033-4042 (2011).
 18. Hamann, L., El-Samalouti, V., Ulmer, A.J., Flad, H.D. & Rietschel, E.T. Components of gut bacteria as immunomodulators. *Int J Food Microbiol* **41**, 141-154 (1998).
 19. Ismail, B. & Nampoothiri, K.M. Molecular characterization of an exopolysaccharide from a probiotic *Lactobacillus plantarum* MTCC 9510 and its efficacy to improve the texture of starchy food. *J Food Sci Technol* **51**, 4012-4018 (2014).
 20. Kim, J.E., Kim, S.Y., Lee, K.W. & Lee, H.J. Arginine deiminase originating from *Lactococcus lactis* ssp. *lactis* American Type Culture Collection (ATCC) 7962 induces G1-phase cell-cycle arrest and apoptosis in SNU-1 stomach adenocarcinoma cells. *Br J Nutr* **102**, 1469-1476 (2009).
 21. Wang, S. et al. Whole Peptidoglycan Extracts from the *Lactobacillus paracasei* subsp. *paracasei* M5 Strain Exert Anticancer Activity In Vitro. *Biomed Res Int* **2018**, 2871710 (2018).
 22. Wu, J., Zhang, Y., Ye, L. & Wang, C. The anti-cancer effects and mechanisms of lactic acid bacteria exopolysaccharides in vitro: A review. *Carbohydr*

- Polym* **253**, 117308 (2021).
23. Ayyash, M. et al. Characterization, bioactivities, and rheological properties of exopolysaccharide produced by novel probiotic *Lactobacillus plantarum* C70 isolated from camel milk. *Int J Biol Macromol* **144**, 938–946 (2020).
 24. Li, S. & Shah, N.P. Characterization, Anti-Inflammatory and Antiproliferative Activities of Natural and Sulfonated Exo-Polysaccharides from *Streptococcus thermophilus* ASCC 1275. *J Food Sci* **81**, M1167–1176 (2016).
 25. Wei, Y., Li, F., Li, L., Huang, L. & Li, Q. Genetic and Biochemical Characterization of an Exopolysaccharide With in vitro Antitumoral Activity Produced by *Lactobacillus fermentum* YL-11. *Front Microbiol* **10**, 2898 (2019).
 26. Deepak, V. et al. In vitro evaluation of anticancer properties of exopolysaccharides from *Lactobacillus acidophilus* in colon cancer cell lines. *In Vitro Cell Dev Biol Anim* **52**, 163–173 (2016).
 27. Matta, H. et al. Development and characterization of a novel luciferase based cytotoxicity assay. *Sci Rep* **8**, 199 (2018).
 28. Bersch, K.L. et al. Bacterial Peptidoglycan Fragments Differentially Regulate Innate Immune Signaling. *ACS Cent Sci* **7**, 688–696 (2021).
 29. Sagiv-Barfi, I. et al. Eradication of spontaneous malignancy by local immunotherapy. *Sci Transl Med* **10** (2018).
 30. Cook, D.P., Gysemans, C. & Mathieu, C. *Lactococcus lactis* As a Versatile Vehicle for Tolerogenic Immunotherapy. *Front Immunol* **8**, 1961 (2017).
 31. Molina, M.S. et al. Regulatory Dendritic Cells Induced by Bendamustine Are Associated With Enhanced Flt3 Expression and Alloreactive T-Cell Death. *Front Immunol* **12**, 699128 (2021).
 32. Kazi, J.U. & Rönstrand, L. FMS-like Tyrosine Kinase 3/FLT3: From Basic Science to Clinical Implications. *Physiol Rev* **99**, 1433–1466 (2019).
 33. Agarwal, Y. et al. Intratumorally injected alum-tethered cytokines elicit potent and safer local and systemic anticancer immunity. *Nat Biomed Eng* **6**, 129–143 (2022).
 34. Chen, L. et al. Bacterial cytoplasmic membranes synergistically enhance the antitumor activity of autologous cancer vaccines. *Sci Transl Med* **13** (2021).
 35. Chen, R. Bacterial expression systems for recombinant protein production: *E. coli* and beyond. *Biotechnol Adv* **30**, 1102–1107 (2012).
 36. Peleg, Y. & Unger, T. Resolving bottlenecks for recombinant protein expression in *E. coli*. *Methods Mol Biol* **800**, 173–186 (2012).
 37. Kessenbrock, K., Plaks, V. & Werb, Z. Matrix metalloproteinases: regulators of the tumor microenvironment. *Cell* **141**, 52–67 (2010).

Reviewers' Comments:

Reviewer #1:

Remarks to the Author:

The manuscript entitled "Bifunctional engineered *Lactococcus lactis*-based in situ vaccination elicits systemic tumour regressions" describes *Lactococcus lactis* expressing a fusion protein of Fms-like tyrosine kinase 3 ligand and co-stimulator OX40 ligand (FOLactis).

Although authors complemented many parts, there are still some weakness noted as follow.

1. The number of animal is still low. In particular, rechallenge study in Fig. 7l and m was done with only 5 animals. I don't think that a reliable conclusion of vaccine effect can be made with this small size of study.
2. In the results, some descriptions in the results section are not realistic. For example, Fig 5f authors described that "There was a significantly higher proportion of NK cells in Lactis-containing groups (Lactis, Lactis+FO, FOLactis), suggesting Lactis contributed mostly to the increase of NK cells." However, there is no statistical significance between NS and Lactis. Another example is "Importantly, the average frequency of OVA-tetramer+CD8+ CTLs in the FOLactis group elevated from 0.8% to 1.7% compared to the levels in the NS group." This statement should be carefully reviewed by authors to make sure where the 0.8% was from. One more is Fig. 2f. Lactis and FOLactis has no statistical difference.
3. There is no description in the Results about Fig. 8f-h.
4. Abbreviation like Flt3L, OX40, APC comes up without description of full name.
5. Bacterial species should be named in italic.
6. At the final paragraph of Introduction, what is "anti-PD1-resistance" meaning?

Reviewer #3:

Remarks to the Author:

Authors have answered all questions and enhanced manuscript quality.

Point-by-point responses to the reviewer

Note: Following are our responses (in blue color) to reviewers' comments and sentences described in the revised manuscript (highlighted in yellow).

Reviewer #1 (Remarks to the Author):

The manuscript entitled “Bifunctional engineered *Lactococcus lactis*-based in situ vaccination elicits systemic tumour regressions” describes *Lactococcus lactis* expressing a fusion protein of Fms-like tyrosine kinase 3 ligand and co-stimulator OX40 ligand (FOLactis).

Although authors complemented many parts, there are still some weaknesses noted as follow.

1. The number of animal is still low. In particular, rechallenge study in Fig. 7l and m was done with only 5 animals. I don't think that a reliable conclusion of vaccine effect can be made with this small size of study.

Answer 1:

Thank you very much for your comments. We designed the experiment with the corresponding depleting antibodies mainly to explore the role of different immune cells in FOLactis-mediated tumour suppression in the subcutaneous CT26 tumour models. Antibody-mediated depletions revealed that CD8⁺ cells and NK cells were critical to tumour control and long-term immunity, while CD4⁺ cells and macrophages played lesser roles on therapeutic efficacy (Fig. 4h-j). The experimental results have shown significant differences among these groups, though there are 5 animals in each group. Other similar studies also have done with 5 animals in each group when performing the antibody depletion experiment^{1, 2}. We hope our explanations can answer your concerns.

2. In the results, some descriptions in the results section are not realistic. For example, (1) Fig 5f authors described that “There was a significantly higher proportion of NK

cells in Lactis-containing groups (Lactis, Lactis+FO, FOLactis), suggesting Lactis contributed mostly to the increase of NK cells.” However, there is no statistical significance between NS and Lactis.

(2) Another example is “Importantly, the average frequency of OVA-tetramer⁺CD8⁺ CTLs in the FOLactis group elevated from 0.8% to 1.7% compared to the levels in the NS group.” This statement should be carefully reviewed by authors to make sure where the 0.8% was from.

(3) One more is Fig. 2f. Lactis and FOLactis has no statistical difference.

Answer 2:

Thank you for pointing out the unclear descriptions.

(1) Natural killer (NK) cells play an important role in innate immunity³. Probiotics such as Lactic acid bacteria have the ability to activate NK cells⁴⁻⁶. Though there is no statistical significance between NS and Lactis, the average proportion of NK cells in the Lactis group is higher than that of NS group. However, the individual difference of mice in the Lactis group is apparent, which may lead to no statistical significance between NS and Lactis. To be more solid and scientific based on the current data, we will revise the description as follows: “There was a significantly higher proportion of NK cells in the Lactis+FO group and FOLactis group compared to the NS group, suggesting that Lactis combined with FO contributed to the increase of NK cells.”

(2) The recognition of tumour neoantigens by the immune system is a key event in the success of immunotherapy in oncology⁷. And about 1% of these “foreign” neoantigens in cancer cells are spontaneously presented to the immune system⁸. Therefore, there might be few antigen-specific T cells in the tumour-bearing mice even without treatment. Previous studies evaluating Tetramer⁺ T cells have also demonstrated similar results^{9, 10}. But the number of antigen-specific T cells is insufficient, so we use FOLactis to modulate the tumour microenvironment to increase their number.

(3) CD69 and CD25 are the respective early and late activation markers of T cells, respectively. The results showed that live Lactis still played important roles in the early activation of T cells. Since FOLactis released FO in the medium continuously,

FOLactis would lead to more sustained activation of T cells (Fig. 2f). We will revise the description as follows: “After treated with live Lactis or FOLactis for 24 hours in vitro, both Lactis and FOLactis induced a higher expression of CD69 (early activation marker) than that in the NS group. We also observed a higher expression of CD25 (late activation marker) in the FOLactis group than that in the Lactis group, leading to the robust generation of IFN- γ (Fig. 2e-f).”

3. There is no description in the Results about Fig. 8f-h.

Answer 3:

The description in the results about Fig. 8f-h is as follows: “In addition to the CT26 tumour models, which tended to be comparatively responsive to immunotherapy, we also demonstrated the universal applicability of this ISV in two ‘cold’ tumour models, the B16F10-OVA melanoma and 4T1 breast tumour mouse models. In alignment with the results in CT26 tumour-bearing BALB/c mice, the ISV with FOLactis exhibited the best efficacy in tumour suppression and survival compared to other groups (Fig. 8a-c, f-h).”

4. Abbreviation like Flt3L, OX40, APC comes up without description of full name.

Answer 4:

Thanks so much for your careful comments. We will revise the description in the manuscript and check the whole manuscript.

5. Bacterial species should be named in italic.

Answer 5:

We appreciate it a lot for your professional and detailed comments. We will revise the description in the manuscript and check the whole manuscript.

6. At the final paragraph of Introduction, what is “anti-PD1-resistance” meaning?

Answer 6:

Anti-PD1-resistance means “not sensitive to anti-PD1 treatment”. The biological and

functional heterogeneity between tumours-both across and within cancer types-poses a challenge for immunotherapy¹¹. Different types of tumours have different immunotherapy sensitivity, which is associated with features of the tumour microenvironment (TME). The TME could be simply characterized into cold (non T cell inflamed) or hot (T cell inflamed)¹². Those so-called hot tumours are characterized by T cell infiltration and molecular signatures of immune activation, whereas cold tumours show striking features of T cell absence or exclusion¹³. The cold tumours present lower response rates to immunotherapy, such as anti-programmed death ligand (PD-L)1/PD-1therapy¹⁴. It is reported that 4T1 and B16F10, which lack tumour infiltrating lymphocytes (TIL), are well-known cold tumours, and PD-1/PD-L1 antibody alone has almost no inhibitory effect on these tumours^{15,16}. To explore the ability of FOlactis to reprogram the tumour immune microenvironment, we used multiple subcutaneous and orthotopic mouse models with poor immune cell infiltration and anti-PD1-resistance.

1. Agarwal, Y. et al. Intratumorally injected alum-tethered cytokines elicit potent and safer local and systemic anticancer immunity. *Nat Biomed Eng* **6**, 129-143 (2022).
2. Chen, L. et al. Bacterial cytoplasmic membranes synergistically enhance the antitumor activity of autologous cancer vaccines. *Sci Transl Med* **13** (2021).
3. Spitzer, M.H. et al. Systemic Immunity Is Required for Effective Cancer Immunotherapy. *Cell* **168**, 487-502. e415 (2017).
4. Garbacz, K. Anticancer activity of lactic acid bacteria. *Semin Cancer Biol* (2022).
5. Miller, L.E., Lehtoranta, L. & Lehtinen, M.J. The Effect of Bifidobacterium animalis ssp. lactis HN019 on Cellular Immune Function in Healthy Elderly Subjects: Systematic Review and Meta-Analysis. *Nutrients* **9** (2017).
6. Ashraf, R. & Shah, N.P. Immune system stimulation by probiotic microorganisms. *Crit Rev Food Sci Nutr* **54**, 938-956 (2014).
7. Yarchoan, M., Johnson, B.A., 3rd, Lutz, E.R., Laheru, D.A. & Jaffee, E.M. Targeting neoantigens to augment antitumour immunity. *Nat Rev Cancer* **17**, 209-222 (2017).
8. Tran, E. et al. Immunogenicity of somatic mutations in human gastrointestinal cancers. *Science* **350**, 1387-1390 (2015).
9. Cheng, K. et al. Bioengineered bacteria-derived outer membrane vesicles as a versatile antigen display platform for tumor vaccination via Plug-and-Display technology. *Nat Commun* **12**, 2041 (2021).

10. Takeshima, T. et al. Local radiation therapy inhibits tumor growth through the generation of tumor-specific CTL: its potentiation by combination with Th1 cell therapy. *Cancer Res* **70**, 2697-2706 (2010).
11. Li, J. et al. Tumor Cell-Intrinsic Factors Underlie Heterogeneity of Immune Cell Infiltration and Response to Immunotherapy. *Immunity* **49**, 178-193.e177 (2018).
12. Gajewski, T.F. The Next Hurdle in Cancer Immunotherapy: Overcoming the Non-T-Cell-Inflamed Tumor Microenvironment. *Semin Oncol* **42**, 663-671 (2015).
13. Duan, Q., Zhang, H., Zheng, J. & Zhang, L. Turning Cold into Hot: Firing up the Tumor Microenvironment. *Trends Cancer* **6**, 605-618 (2020).
14. Zemek, R.M. et al. Sensitization to immune checkpoint blockade through activation of a STAT1/NK axis in the tumor microenvironment. *Sci Transl Med* **11** (2019).
15. Wu, Y. et al. ARIH1 signaling promotes anti-tumor immunity by targeting PD-L1 for proteasomal degradation. *Nat Commun* **12**, 2346 (2021).
16. Huang, L. et al. Mild photothermal therapy potentiates anti-PD-L1 treatment for immunologically cold tumors via an all-in-one and all-in-control strategy. *Nat Commun* **10**, 4871 (2019).